# Synthetic control over the binding configuration of luminescent *sp*³-defects in single-walled carbon nanotubes

Simon Settele [1], Felix J. Berger [1,2], Sebastian Lindenthal[1], Shen Zhao[3], Abdurrahman Ali El Yumin[1,2], Nicolas F. Zorn [1,2], Andika Asyuda [1], Michael Zharnikov [1], Alexander Högele[3,4] & Jana Zaumseil [1,2 ✉]

The controlled functionalization of single-walled carbon nanotubes with luminescent *sp*³-defects has created the potential to employ them as quantum-light sources in the near-infrared. For that, it is crucial to control their spectral diversity. The emission wavelength is determined by the binding configuration of the defects rather than the molecular structure of the attached groups. However, current functionalization methods produce a variety of binding configurations and thus emission wavelengths. We introduce a simple reaction protocol for the creation of only one type of luminescent defect in polymer-sorted (6,5) nanotubes, which is more red-shifted and exhibits longer photoluminescence lifetimes than the commonly obtained binding configurations. We demonstrate single-photon emission at room temperature and expand this functionalization to other polymer-wrapped nanotubes with emission further in the near-infrared. As the selectivity of the reaction with various aniline derivatives depends on the presence of an organic base we propose nucleophilic addition as the reaction mechanism.

¹ Institute for Physical Chemistry, Universität Heidelberg, Heidelberg, Germany. ² Centre for Advanced Materials, Universität Heidelberg, Heidelberg, Germany. ³ Fakultät für Physik, Munich Quantum Center and Center for NanoScience (CeNS), Ludwig-Maximilians-Universität München, München, Germany. ⁴ Munich Center for Quantum Science and Technology (MCQST), München, Germany. ✉email: zaumseil@uni-heidelberg.de

Covalent functionalization of single-walled carbon nanotubes (SWNTs) has a long history and many different types of chemistry have been explored[1–3]. However, a major paradigm shift occurred with the discovery of luminescent $sp^3$ defects[4–10]. Rather than quenching near-infrared photoluminescence (PL) of semiconducting SWNTs, these $sp^3$ defects with covalently attached functional groups were found to create highly emissive states at lower energies. Mobile excitons, which result in $E_{11}$ emission if not quenched[11], are funneled to these defect states with long PL lifetimes and thus the usually low PL quantum yield (PLQY) of SWNTs increases significantly[4,12,13]. Luminescent defects (or organic color centers)[7] have expanded the emission range of SWNTs further toward the near-infrared and opened up applications in sensing[14,15], in vivo imaging within the second biological window[16,17], and as quantum-light sources with room-temperature single-photon emission[18–22]. Various synthetic methods (e.g., radical-based reactions with aryl diazonium salts) allow for different functional groups to be attached to the SWNTs, which lead to additional small energetic shifts[18,23–26]. However, defect emission in general occurs over a wide spectral range (e.g., from 1100 to 1350 nm for (6,5) SWNTs). Indeed, the emission wavelength of $sp^3$-functionalized SWNTs is less determined by the nature of the attached functional group than by the precise defect-binding configuration within the $sp^2$-hybridized SWNT lattice[20,24]. For any functional group that forms a bond with the nanotube, thus creating one $sp^3$ carbon, another carbon atom, either in ortho or para position to the first, must be saturated (e.g., with another substituent, -OH or -H group). For chiral nanotubes, this requirement leads to six possible binding configurations. Each of these is associated with a distinct defect energy and thus emission wavelength as corroborated by density functional theory calculations[9,27,28]. Most functionalization methods produce a mix of configurations and thus defect emission bands, the most common being the $E_{11}^\star$ and the even more red-shifted $E_{11}^{\star-}$ emission with a longer PL lifetime[29–31]. They are considered to originate from the ortho-$L_{90}$ (ortho++, circumferential direction) and ortho-$L_{30}$ (ortho+, orientation along the nanotube axis) configurations of the two $sp^3$ carbons, respectively[20,24,27]. Spectral variation is a problem for practical applications and must be reduced to ideally one single type of defect emission with narrow wavelength and PL lifetime distributions for the entire ensemble of functionalized nanotubes.

The more red-shifted $E_{11}^{\star-}$ emission is closer to telecommunication wavelengths and is better suited for high-purity single-photon emission at room temperature. However, it is usually found for only a small portion of functionalized nanotubes. Saha et al.[20] suggested the use of achiral zigzag-SWNTs whose symmetry leads to degenerate defect configurations and hence single peak emission. However, zigzag nanotubes are a very rare species in typical nanotube raw materials[32] and thus difficult to obtain and purify in significant amounts. Another approach is the functionalization of SWNTs with either divalent functional groups (e.g., >CF$_2$)[29] or bidentate reactants with short bridging moieties (e.g., bisdiazonium compounds)[31], which increase the probability of a specific binding configuration, albeit with limitations. Gaining real synthetic control over the defect-binding configuration and hence emission wavelength in a simple, reproducible, and possibly scalable manner is highly desirable.

Here we demonstrate a synthetic approach for the introduction of luminescent $sp^3$ defects in carbon nanotubes, with which we can either create both types of binding configurations leading to $E_{11}^\star$ and $E_{11}^{\star-}$ emission, or exclusively the configuration for $E_{11}^{\star-}$ emission at longer wavelengths and with longer PL lifetimes. We employ polymer-sorted monochiral (6,5) SWNTs for this functionalization[33], as they are easily purified and are available in relatively large amounts. The use of polymer-wrapped SWNTs in organic solvents avoids the limitations of reactions in aqueous dispersions[10] and, hence, dramatically expands the available chemical toolbox. The selectivity and reactivity of the functionalization step is controlled by the concentration of potassium *tert*-butoxide (KO$^t$Bu) as a base and, thus, we propose nucleophilic addition as the underlying reaction mechanism as opposed to the commonly applied radical-based functionalization routes that mainly produce defects for $E_{11}^\star$ emission.

## Results

**Synthetic control over defect emission from (6,5) SWNTs.** Polymer-wrapped (6,5) SWNTs (length 1–2 μm) serve as the model system for the controlled introduction of different luminescent $sp^3$ defects in this study and were selectively dispersed and purified by shear-force mixing in toluene with a polyfluorene copolymer (poly[(9,9-dioctylfluorenyl-2,7-diyl)-*alt*-(6,6')-(2,2'-bipyridine)] (PFO-BPy), see Fig. 1a and "Methods"). Excess polymer was removed by vacuum filtration and all reactions were performed with (6,5) nanotubes redispersed in fresh toluene (Supplementary Fig. 1). Functionalization occurred via base-mediated coupling with 2-haloanilines (2-iodoaniline if not specified otherwise) as shown in Fig. 1a, either under ultraviolet (UV)-light irradiation (~365 nm, blue reaction path) or in the dark (red reaction path). As we will see, the key component for controlling the reaction is the organic base KO$^t$Bu. Tetrahydrofuran (THF, 8.3 vol%) and dimethyl sulfoxide (DMSO, 8.3 vol%) were added as co-solvents to increase reactivity and selectivity. The functionalization was performed in an open flask at room temperature (10–180 min). Vacuum filtration of the mixture quenched the reaction by separating the reagents from the nanotubes. The collected SWNTs were washed with methanol and redispersed again in toluene for characterization. For a detailed step-by-step description, see Supplementary Methods 1 and Supplementary Table 1.

Nanotubes that were functionalized with 2-iodoaniline under UV illumination exhibited PL spectra (Fig. 1b) with the $E_{11}$ emission of the mobile excitons and two red-shifted emission bands at ~1130 nm and ~1250 nm, which we assign to $E_{11}^\star$ and $E_{11}^{\star-}$, respectively. The relative intensities of the two defect emission bands could be controlled by the amount of KO$^t$Bu (0.5–3 eq.) in the reaction mixture, whereas the concentration of 2-iodoaniline was kept constant. With increasing base concentration, the reactivity increased (i.e., higher ratio of defect to $E_{11}$ intensity) and the $E_{11}^{\star-}$ emission became more and more dominant. Evidently, a higher base concentration favors the creation of the defect-binding configuration corresponding to $E_{11}^{\star-}$. Nevertheless, the defect configuration for $E_{11}^\star$, which is the majority product for reactions with diazonium salts[34] as well as photoactivated reactions of 4-iodoaniline in aqueous dispersion[23], is still significant at lower base concentrations as long as UV illumination is applied.

Surprisingly, when the same reaction was performed in the dark (i.e., without UV-light activation), only the more red-shifted $E_{11}^{\star-}$ emission feature (peak width 65 meV) was observed (Fig. 1c). The emission intensity and, thus, the number of defects increased steadily with base concentration and reaction time, and could easily be tuned using these parameters (Fig. 1c, d). Even after very long reaction times, no other defect emission bands appeared. Concentration-dependent functionalization reactions revealed that the highest reactivity and selectivity for defects with $E_{11}^{\star-}$ emission is achieved when the aniline reagent is used in excess compared to the SWNTs (see Supplementary Note 1 and Supplementary Fig. 2). To the best of our knowledge, this reaction constitutes the first example of $sp^3$ functionalization of chiral carbon nanotubes that exhibits such a high selectivity for this

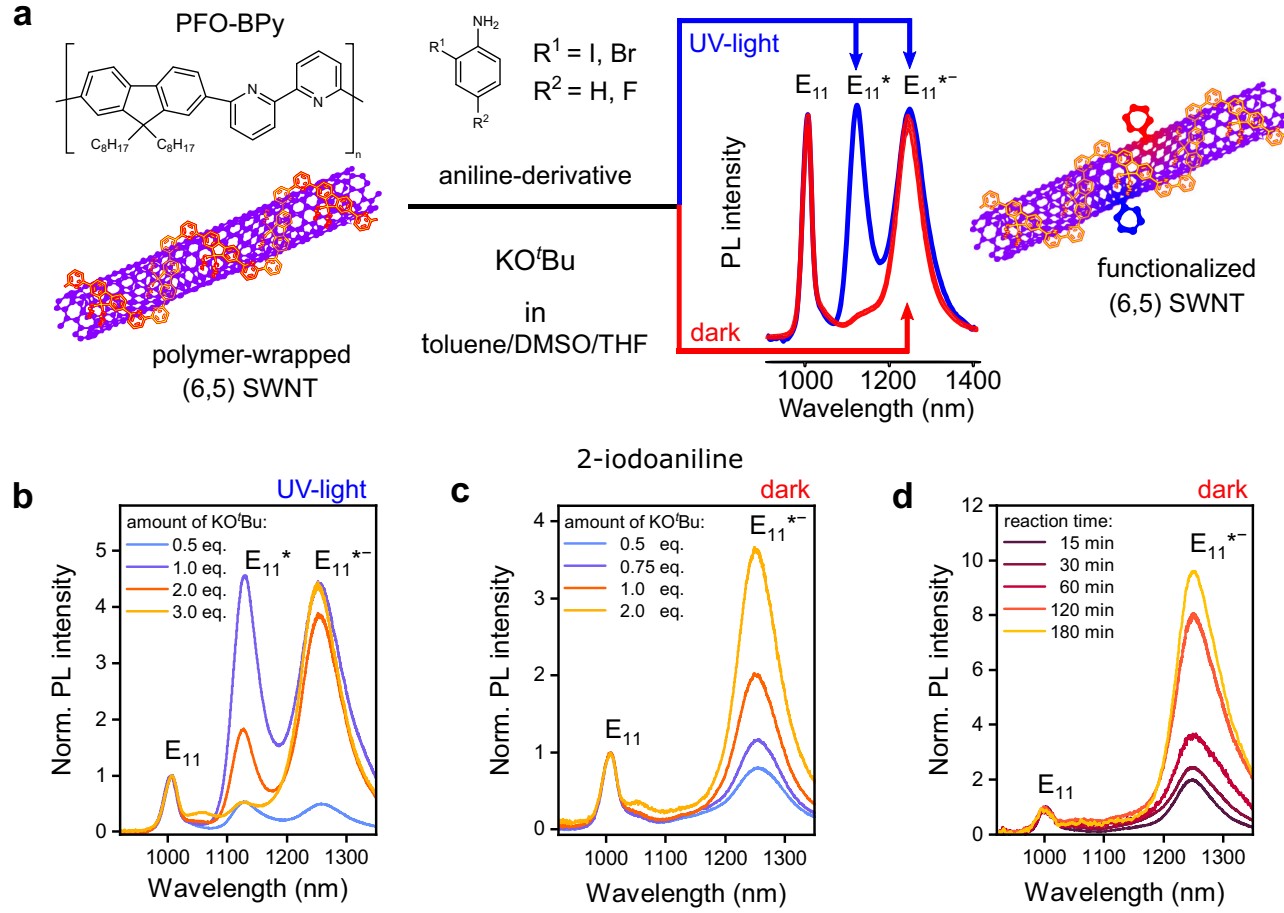

**Fig. 1 Controlling $sp^3$-defect emission from (6,5) SWNTs. a** Scheme for functionalization of PFO-BPy-wrapped (6,5) SWNTs in toluene/DMSO/THF with aniline derivatives and base KO$^t$Bu under UV-light irradiation (blue) and in the dark (red), respectively. Red-shifted defect emission bands are labeled as $E_{11}^*$ (~1130 nm) and $E_{11}^{*-}$ (~1250 nm). **b, c** PL spectra of (6,5) SWNTs functionalized with 2-iodoaniline and varying concentrations of KO$^t$Bu under UV-light irradiation (**b**) and in the dark (**c**); reaction time 30 min. **d** Evolution of $E_{11}^{*-}$ emission normalized to $E_{11}$ intensity for increasing reaction times with 2 eq. of KO$^t$Bu in the dark. The concentration of 2-iodoaniline was kept at 29.30 mmol L$^{-1}$.

specific type of defect-binding configuration and exclusively leads to strongly red-shifted emission ($E_{11}^{*-}$).

The established reaction scheme was not limited to 2-iodoaniline but was also performed successfully with 2-bromoaniline and 5-fluoro-2-iodoaniline (see Supplementary Fig. 3), although with different reactivities. A high selectivity and yield for $E_{11}^{*-}$ defects were achieved with 2-bromoaniline, whereas the additional steric repulsion in 5-fluoro-2-iodoaniline slightly lowered reactivity and selectivity. Overall, the base-mediated functionalization of nanotubes is selective, scalable, and uses inexpensive reagents that are easy and safe to handle. By employing this procedure, we were able to obtain high-quality dispersions of functionalized (6,5) SWNTs with only $E_{11}^{*-}$ defects, which previously were only accessible in conjunction with $E_{11}^*$ defects. Before investigating the origin of this selectivity, the spectroscopic properties of these functionalized nanotubes and their possible application as single-photon emitters will be discussed and demonstrated.

**Spectroscopic properties of functionalized (6,5) SWNTs.** As shown above (Fig. 1d), the reaction time represents an excellent tool to precisely tune the $E_{11}^{*-}$ defect density of polymer-wrapped (6,5) SWNTs functionalized with 2-iodoaniline in the dark, which is reflected in the linear correlation of the $E_{11}^{*-}/E_{11}$ PL intensity

ratio (Fig. 2a). Various additional metrics can be used to quantify the degree of functionalization. These are the integrated $E_{11}^{*-}/E_{11}$ absorption peak ratio and the Raman D/G$^+$-mode ratio. Both are expected to be directly proportional to the number of $sp^3$ defects[35]. The $E_{11}^{*-}$ defect absorption band, located around ~1247 nm (Fig. 2b), is much weaker than the $E_{11}$ absorption but increases steadily with reaction time. The integrated $E_{11}^{*-}/E_{11}$ absorption ratios for different reaction times also reveal a linear correlation with the $E_{11}^{*-}/E_{11}$ PL intensity ratios (Fig. 2c). Their large difference (approximately a factor of 100) indicates the strong funneling effect of mobile excitons toward the luminescent defect states. The Stokes shift between $E_{11}^{*-}$ absorption and emission was rather small with 9–16 meV, similar to values for $E_{11}^*$ defects introduced by diazonium chemistry[34]. The systematic increase of the Raman D-mode signal and thus D/G$^+$ mode ratio with reaction time also showed a linear correlation with the $E_{11}^{*-}/E_{11}$ emission ratio (Supplementary Fig. 4) and can be used as an independent metric for the defect density.

As previously shown for $E_{11}^*$ defects[4,34], the controlled functionalization of SWNTs can be used to significantly enhance their ensemble PLQYs, albeit only at optimum defect densities. PLQY data (absolute values obtained with an integration sphere, see "Methods") for (6,5) SWNTs with $E_{11}^{*-}$ defects are shown in Fig. 2d. The intensity of the $E_{11}$ emission and thus also its contribution to the total PLQY decreases steeply with the number

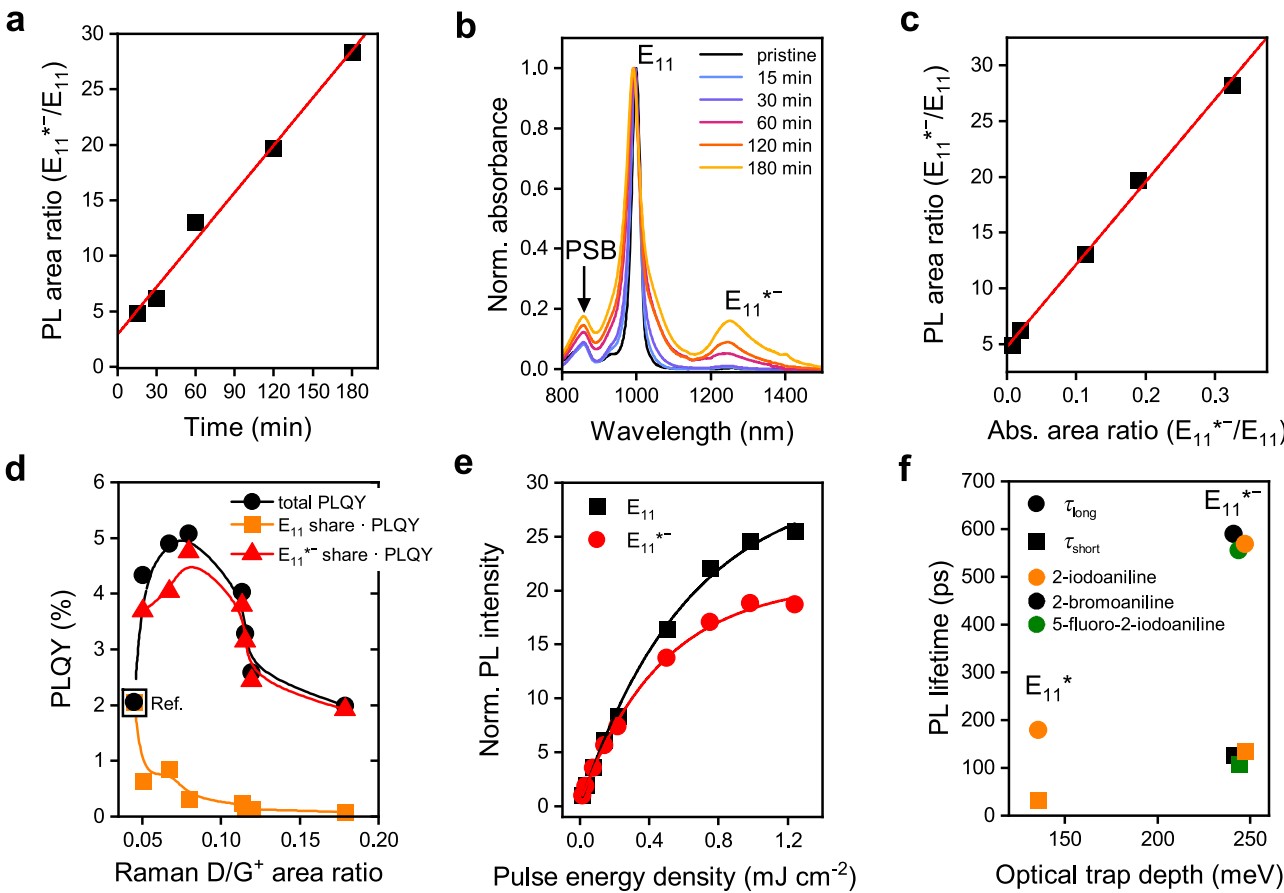

**Fig. 2 Spectroscopic characterization of functionalized (6,5) SWNTs. a–e** (6,5) SWNTs functionalized with 2-iodoaniline in the dark. **a** Integrated $E_{11}{}^{*-}/E_{11}$ emission ratio vs. reaction time with linear fit. **b** Absorption spectra of functionalized (6,5) SWNTs normalized to $E_{11}$ transition, showing phonon sideband (PSB, ~860 nm) and $E_{11}{}^{*-}$ absorption band (~1247 nm) increasing with reaction time. It is noteworthy that a broadening of the $E_{11}{}^{*-}$ defect absorption band may occur due to a scattering background caused by increasing aggregation of SWNTs at very high defect densities. **c** Integrated $E_{11}{}^{*-}/E_{11}$ emission vs. $E_{11}{}^{*-}/E_{11}$ absorbance ratio with linear fit. **d** Photoluminescence quantum yield (PLQY) of functionalized SWNTs (total and separated into spectral shares of $E_{11}{}^{*-}$ and $E_{11}$) vs. integrated Raman $D/G^{+}$ ratios as metric for defect density (lines are guides to the eye). **e** Normalized intensity of $E_{11}{}^{*-}$ (red) and $E_{11}$ (black) emission vs. pulse energy density, indicating faster saturation of defect state than band-edge exciton emission. **f** PL lifetime ($\tau_{long}$, $\tau_{short}$) vs. optical trap depth for $E_{11}{}^{*}$ and $E_{11}{}^{*-}$ defects created by functionalization of (6,5) SWNTs with 2-iodoaniline (orange), 2-bromoaniline (black), and 5-fluoro-2-iodoaniline (green).

of defects (here quantified by the Raman $D/G^{+}$ ratio), whereas the $E_{11}{}^{*-}$ share rises sharply as mobile excitons are trapped by the emissive defects. At even higher defect densities, the absolute $E_{11}$ and $E_{11}{}^{*-}$ emission and thus PLQY decreases again as the $sp^2$ lattice is disrupted by too many $sp^3$ carbons, preventing the formation of extended electronic states. Areas with such high $sp^3$ carbon concentrations act as quenching sites, thus decreasing the overall number of excitons for emission. Nevertheless, a strong brightening effect compared to the pristine samples (PLQY ~ 2%) was observed for the total PLQY of functionalized nanotubes, reaching up to 5.1% at optimal defect density. This represents a significant improvement compared to polymer-wrapped (6,5) SWNTs functionalized with diazonium salts (mainly $E_{11}{}^{*}$ defects) and is consistent with the larger optical trap depth (difference between $E_{11}$ and defect emission energy) of the $E_{11}{}^{*-}$ defects (241–247 meV)[34].

Another characteristic feature of SWNT defect state emission is its nonlinear behavior at high pump power, i.e., the ratio of defect to $E_{11}$ emission depends strongly on the chosen excitation method (lamp or laser, continuous or pulsed) and power (Fig. 2e and Supplementary Fig. 5), and decreases with excitation density. This power dependence can be explained with filling of the long-lived defect states at higher excitation densities[36]. It is more

pronounced for the $E_{11}{}^{*-}$ defects with their emission already saturating at relatively low laser powers compared to the $E_{11}{}^{*}$ defects (see Supplementary Fig. 6).

The increase in total PLQY and the power dependence of emission are best understood within the framework of the defect state relaxation dynamics. Hence, the PL decays for (6,5) SWNTs functionalized with 2-iodoaniline, 2-bromoaniline, or 5-fluoro-2-iodoaniline were recorded using time-correlated single-photon counting (TCSPC) and fitted as biexponential decays (see Supplementary Fig. 7) with a short ($\tau_{short}$) and long ($\tau_{long}$) lifetime component. As shown in Fig. 2f, the longer lifetimes increase significantly with optical trap depth from 179 ps for $E_{11}{}^{*}$ (only for 2-iodoaniline, trap depth 136 meV) to up to 589 ps for $E_{11}{}^{*-}$ (trap depth 241 meV, see also Supplementary Table 2). The values are consistent with previous measurements of $E_{11}{}^{*-}$ defects that were introduced in much smaller numbers by diazonium chemistry[18,37]. The temperature dependence of the defect emission and thermal detrapping of excitons are presented and discussed in Supplementary Note 2 and Supplementary Figs. 8–11.

**Single-photon emission and individual SWNT spectra.** One important application of luminescent $sp^3$ defects in SWNTs is as

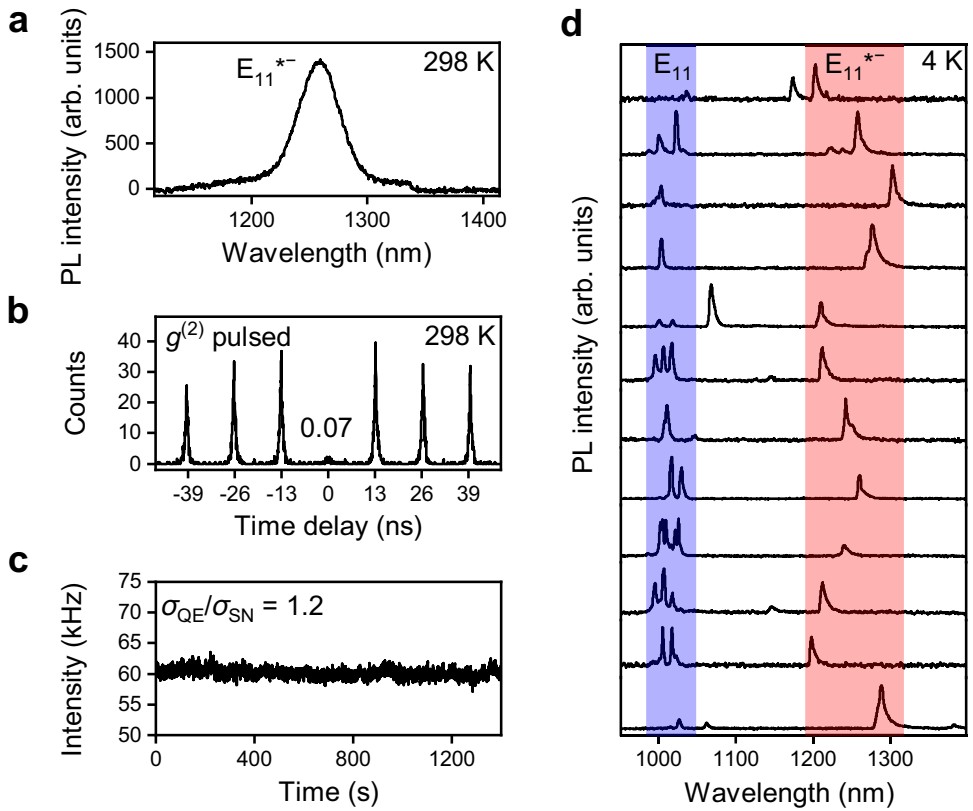

**Fig. 3 PL characteristics of individual (6,5) SWNTs with $E_{11}^{\star-}$ defects. a** PL spectrum of a single (6,5) SWNT functionalized with 2-iodoaniline embedded in a polystyrene matrix at room temperature (298 K) collected under continuous wave excitation at the $E_{11}$ transition. **b** Second-order photon-correlation function $g^{(2)}(t)$ of $E_{11}^{\star-}$ emission at room temperature, confirming high single-photon purity (93%). **c** PL time trace showing a count rate fluctuation near the shot-noise limit. The deviation of the PL intensity distribution of the emitter ($\sigma_{QE}$) from the Poisson distribution of the detector shot noise ($\sigma_{SN}$) is expressed as $\sigma_{QE}/\sigma_{SN} = ((\langle n^2 \rangle - \langle n \rangle^2)/\langle n \rangle)^{0.5}$, with $n$ being the number of PL counts within a 0.1 s timeframe. **d** PL spectra of 12 individual (6,5) SWNTs functionalized with 2-iodoaniline and embedded in a polystyrene matrix recorded at 4 K. The spectral ranges of the $E_{11}$ (blue) and $E_{11}^{\star-}$ (red) emission peaks are highlighted.

quantum-light sources in the near-infrared and at room temperature[18,38]. Defects with large optical trap depths are best suited for single-photon emission at higher temperatures and, ideally, emission should be within one of the standard telecommunication bands (O-band or C-band). The selectively produced $E_{11}^{\star-}$ defects in polymer-wrapped SNWTs fullfil a good part of these requirements. To confirm their suitability as single-photon emitters, Hanbury–Brown–Twiss experiments were performed on individualized (6,5) SWNTs with a low concentration of $E_{11}^{\star-}$ defects embedded in a polystyrene matrix at room temperature (see Fig. 3a–c). The probability of two consecutive photon detection events is expressed in the second-order photon-correlation function $g^{(2)}(t)$. Under pulsed excitation, the absence of detection events at zero time delay (here, $g^{(2)}(0) = 0.07$) indicates photon antibunching with high single-photon purity (93%) for the $E_{11}^{\star-}$ defects. Furthermore, high count rates (~60 kHz) were observed during the measurement with fluctuations near the shot-noise limit.

Comparable values were found by He et al.[18] at similar emission wavelengths (1280 nm) for polymer-wrapped (6,5) SWNTs functionalized with a diazonium salt. However, in that case, the defect emission of the whole set of individual (6,5) SWNTs was distributed over a large spectral range (1000–1350 nm) due to the presence of many nanotubes with $E_{11}^{\star}$ defects and only few with $E_{11}^{\star-}$ defects in the ensemble[24]. It is noteworthy that for the application of SWNTs as single-photon sources, homogeneous emission line broadening also represents a significant challenge. However, cavity enhancement of the radiative emission rate via

the Purcell effect was recently shown to improve the photon indistinguishability[22]. Here, the exclusive introduction of $E_{11}^{\star-}$ defects via the reaction with 2-iodoaniline in the dark enables a better matching with the limited bandwith of such cavities as the distribution of emission wavelengths is reduced to 1200–1300 nm.

The reduced spectral diversity is also reflected at the single-nanotube level as revealed by PL spectra of individualized (6,5) SWNTs with a low $E_{11}^{\star-}$ defect density embedded in a polystyrene matrix at 4 K (Fig. 3d). Almost all defect emission peaks appear within an 80 meV window (highlighted in red), with peak widths between 2.9 and 5 meV (limited by a low-resolution grating) and exhibiting the typical asymmetric lineshapes of polymer-wrapped nanotubes. The remaining energetic shifts between emission peaks are likely due to variations in the dielectric environment[18]. It is noteworthy that the segmentation of the $E_{11}$ emission (highlighted in blue) at 4 K originates from random localizations of excitons in long polymer-wrapped SWNTs or from other extrinsic defects introduced during initial processing. They are also present in SWNTs that were not specifically functionalized[39–41]. An integrated spectrum of 62 spots closely resembles the ensemble spectrum of a thin film at room temperature (Supplementary Fig. 12).

**Luminescent $sp^3$ defects in larger diameter SWNTs.** Although the $E_{11}^{\star-}$ emission from (6,5) nanotubes (~1250 nm) is already close to relevant telecommunication bands and is well located within the second biological window (1000–1350 nm), further red-shifted PL is desirable. This might be achieved by introducing

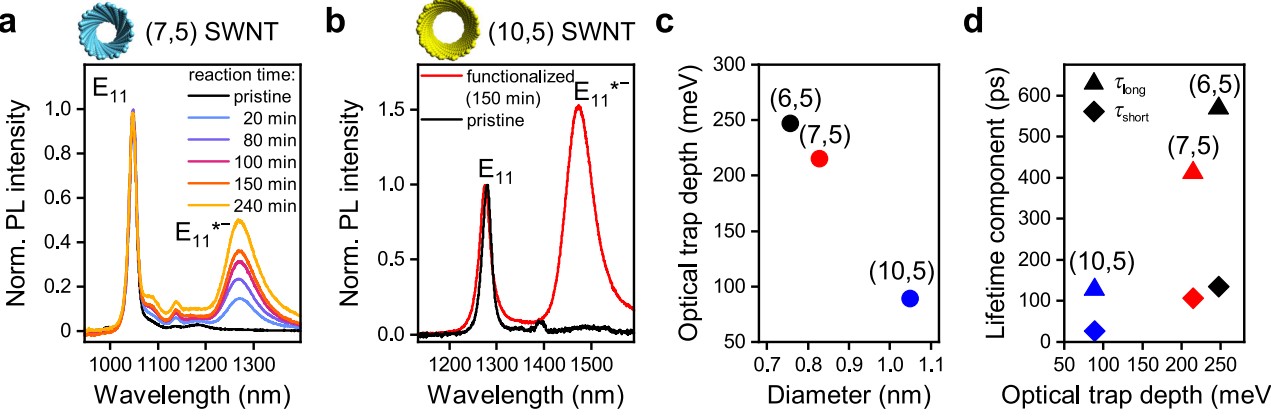

**Fig. 4 Functionalization of polymer-wrapped SWNTs with larger diameters. a** PL spectra of PFO-wrapped (7,5) SWNTs functionalized with 2-iodoaniline in the dark after different reaction times with a red-shifted $E_{11}{}^{\star-}$ emission band (~1270 nm). **b** PL spectra of F8BT-wrapped pristine (10,5) SWNTs and functionalized with 2-iodoaniline in the dark for 150 min. Functionalization gives rise to a red-shifted $E_{11}{}^{\star-}$ emission band (~1474 nm). The concentration of 2-iodoaniline was kept at 29.30 mmol L$^{-1}$. **c** Optical trap depths of $E_{11}{}^{\star-}$ defect states as a function of SWNT diameter. **d** $E_{11}{}^{\star-}$ defect fluorescence lifetimes (long and short) depending on optical trap depth.

the same $E_{11}{}^{\star-}$ defects in polymer-wrapped semiconducting nanotubes with larger diameters. Two possible candidates are (7,5) SWNTs (diameter 0.83 nm, $E_{11}$ transition at 1047 nm) and (10,5) SWNTs (diameter 1.05 nm, $E_{11}$ transition at 1276 nm). They can be selectively dispersed in toluene using suitable polyfluorene derivatives (poly[(9,9-dioctylfluorenyl-2,7-diyl)] (PFO) and poly[(9,9-dioctylfluorenyl-2,7-diyl)-alt-(1,4-benzo [2,1,3]thiadiazole)] (F8BT), see "Methods" and Supplementary Fig. 13). Nearly monochiral dispersions were functionalized with 2-iodoaniline in the dark. The resulting PL spectra are displayed in Fig. 4a, b, respectively. In both cases, almost only the $E_{11}{}^{\star-}$ defect emission was observed; however, the achievable concentration of defects remained relatively low in comparison to the (6,5) SWNTs. Very long reaction times were required to reach moderate $E_{11}{}^{\star-}/E_{11}$ PL ratios. The Raman D/G$^+$ ratios for (7,5) nanotubes were also significantly lower than those for (6,5) SWNTs (Supplementary Fig. 14).

The $E_{11}$ and $E_{11}{}^{\star-}$ emissions from SWNTs with larger diameters occur at longer wavelengths. However, the defect emission bands for (7,5) and (10,5) SWNTs appear at 1270 nm and 1474 nm, respectively, and thus not as far red-shifted as desired for applications. The optical trap depth of the $E_{11}{}^{\star}$ defects was previously shown to monotonically decrease with increasing nanotube diameter, making even more red-shifted emission increasingly harder to achieve[42]. This correlation was also found for the $E_{11}{}^{\star-}$ defects in (7,5) and (10,5) SWNTs (optical trap depths of 215 meV and 89 meV, respectively) as shown in Fig. 4c. Shallower trap depths also lead to shorter defect PL lifetimes (Fig. 4d) reaching only 128 ps for the long lifetime component of $E_{11}{}^{\star-}$ defects in (10,5) nanotubes. Both effects make trapping and radiative recombination at these defect sites less efficient and no increase in PLQY was observed upon functionalization as shown for (7,5) nanotubes with various defect densities (Supplementary Fig. 15).

**Functionalization mechanism and origin of selectivity.** The selectivity of the reaction of 2-haloanilines with SWNTs for defect-binding configurations with either $E_{11}{}^{\star}$ or $E_{11}{}^{\star-}$ emission depends on conditions such as UV-light illumination and KO$^t$Bu concentration. The different selectivities suggest divergent reaction pathways that are explored in more detail in the following. Previous functionalization reactions of nanotubes with anilines and haloanilines employed UV activation of the reagent or the nanotubes and produced either only $E_{11}{}^{\star}$ emission or both $E_{11}{}^{\star-}$

and $E_{11}{}^{\star-}$ emission[23,25,30]. It is commonly assumed that the halogen is eliminated in the process of activation of haloanilines (dehalogenation). However, under the reaction conditions employed here, i.e., with KO$^t$Bu as the base and in the dark, a high reactivity and selectivity was observed not only for reactions with 2-iodoaniline and 2-bromoaniline but also for 2-fluoroaniline (see Fig. 5a). In the latter case, the reaction is not expected to occur via a dehalogenation process, as the C–F bond is one of the strongest known carbon bonds.

To answer the question what chemical group is actually attached to the nanotubes after functionalization, we performed X-ray photoemission spectroscopy (XPS) measurements on thin films of (6,5) SWNTs functionalized with 2-fluoroaniline, using the strong signal of the F 1s core level as a metric (see Fig. 5b). Clearly, the fluorine is present after functionalization, indicating that dehalogenation does not take place. The extracted binding energy (689.5 eV) is in agreement with literature values of alkyl and aryl C–F bonds[43,44]. Additional experiments were carried out with 5-fluoro-2-iodoaniline (Supplementary Fig. 16), with similar results. It is noteworthy that it was not possible to employ thermogravimetric analysis coupled with mass spectrometry[45] to further confirm the nature of attached functional groups due to the necessarily low defect density and limited amount of purified (6,5) SWNTs.

To understand the reaction mechanism of (6,5) SWNTs in the presence of KO$^t$Bu and in the dark, we performed a large number of reference and control experiments (see Supplementary Tables 3 and 4). Surprisingly, functionalization of (6,5) nanotubes and strong defect emission bands ($E_{11}{}^{\star}$ and $E_{11}{}^{\star-}$) even occurred without any aniline derivative, thus suggesting side reactions that create the same defect-binding configuration. Control reactions were conducted both in toluene and THF. These experiments revealed a complex influence of reaction parameters and only the key aspects will be discussed here (see Supplementary Note 3 for details). Briefly, $E_{11}{}^{\star-}$ emission bands only appear in the presence of KO$^t$Bu and a higher selectivity is achieved with the addition of DMSO as a co-solvent. Thus, the introduction of $E_{11}{}^{\star-}$ defects can be associated with the basicity of the system, which is greatly increased by DMSO. In contrast to that, $E_{11}{}^{\star}$ emission bands appear only in the absence of DMSO or under UV irradiation. Irradiation of polymer-wrapped (6,5) SWNTs in THF or toluene with UV light without reagents did not result in any functionalization. Functionalization with 2-iodoaniline in the dark showed the typical temperature

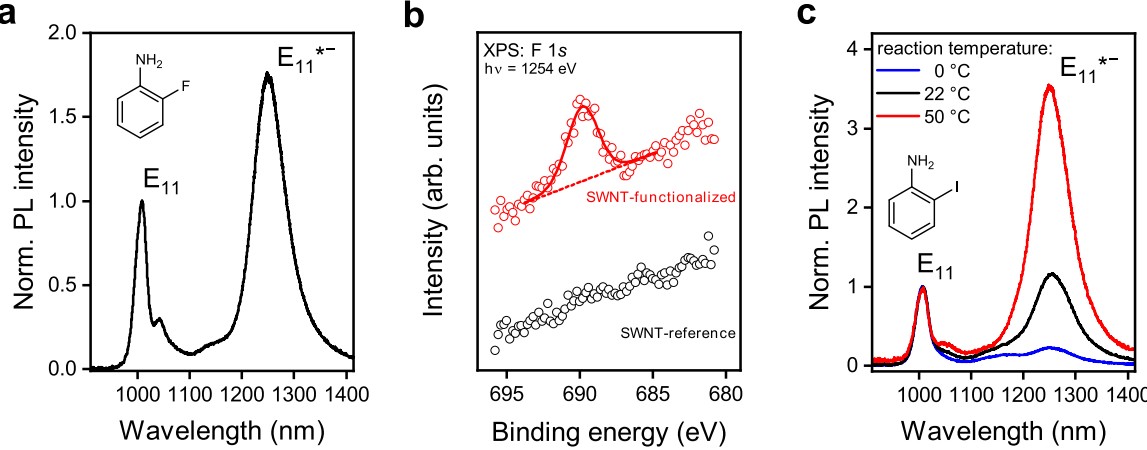

**Fig. 5 Mechanistic investigation of functionalization. a** PL spectrum of (6,5) SWNTs functionalized with 2-fluoroaniline in the dark with a strong defect emission band at 1250 nm. **b** F 1s XPS spectra of (6,5) SWNTs after reaction with 2-fluoroaniline with (red circles including peak fit, red line) and without (black circles; reference) addition of KO$^t$Bu as base. The presence of the F 1s signal indicates covalent functionalization with 2-fluoroaniline, whereas the absence of this signal for the reference sample indicates lack of functionalization. **c** PL spectra of (6,5) SWNTs functionalized with 2-iodoaniline at different reaction temperatures in the dark. The concentrations of 2-fluoroaniline and 2-iodoaniline were kept at 29.30 mmol L$^{-1}$.

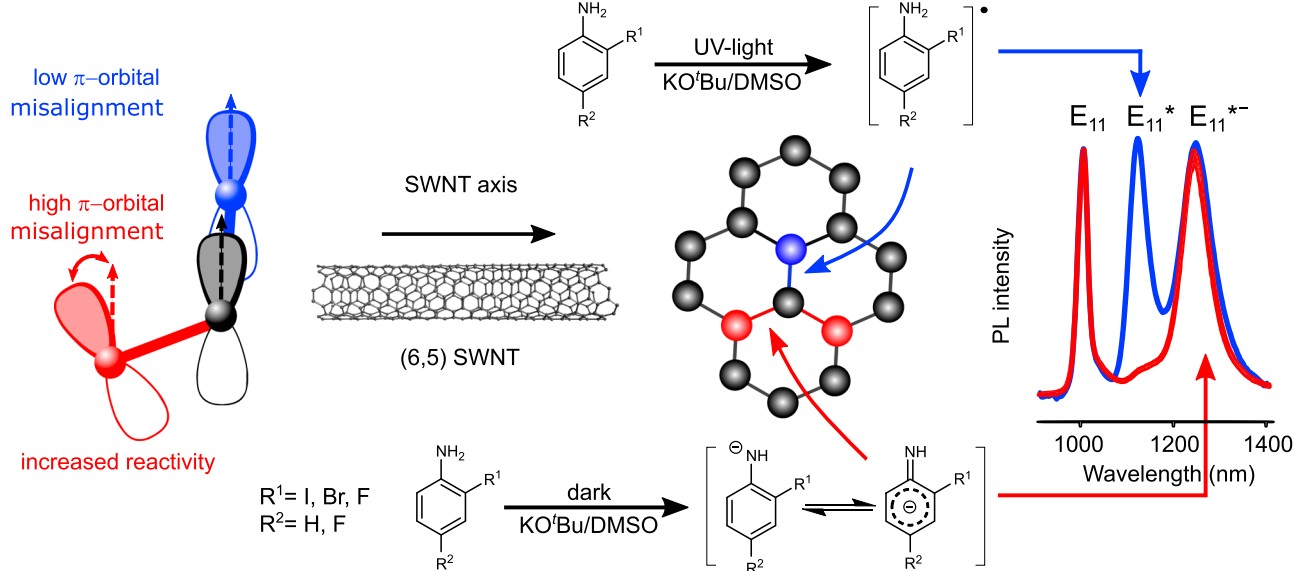

**Fig. 6 Proposed functionalization mechanism for (6,5) SWNTs.** Aniline derivatives can follow two reaction paths in the presence of KO$^t$Bu and DMSO. Under UV-light irradiation, a radical species is formed that attacks the C–C bonds in circumferential direction, leading to the defect configuration for the E$_{11}$* emission band (blue line). When the reaction is performed in the dark, deprotonation of the amine group and charge delocalization occurs. A nucleophilic attack on bonds with large $\pi$-orbital misalignment creates the defect configuration for the E$_{11}$*$^-$ emission band (red line). Similar reaction paths are expected for phenols, thiophenols, and indoles with different carbanion intermediates. It is noteworthy that the deprotonated aniline species is in equilibrium with its protonated form.

dependence of thermally activated reactions (see Fig. 5c) without losing its selectivity for E$_{11}$*$^-$ defects.

Considering the available data, we propose two possible functionalization mechanisms based on radical and nucleophilic reaction paths that determine the final defect-binding configurations. The E$_{11}$* emission band is associated with the ortho-L$_{90}$ (ortho++) configuration[20,24,27], which is oriented in circumferential direction for (6,5) SWNTs (see Fig. 6, blue carbon atom). It is usually observed for reactions with diazonium salts and other monovalent functionalization reactions, e.g., with aryl halides, relying on radical formation. In our case, E$_{11}$* emission mainly appears when the formation of radicals is assisted by UV-light illumination or electron transfer. Consequently, we assign the E$_{11}$* emission to a reaction path where the carbon bonds in

circumferential direction of the SWNT are attacked by radical species (indicated in blue in Fig. 6). Recent studies have shown that functionalization by radicals can also introduce further red-shifted E$_{11}$*$^-$ defects, but only when the already existing defects induce a higher $\pi$-orbital misalignment in bonds along the tube axis[29]. In contrast to that, non-radical reactive species exhibit an intrinsic preference for C–C bonds with large $\pi$-orbital misalignment and thus preferably attack carbon bonds in the axial direction (highlighted in red). Such reactive species are nucleophilic intermediates.

Nucleophilic intermediates can be formed via deprotonation of aniline derivatives by a base and stabilization of the anion through various possible resonance structures[46,47]. This initial deprotonation represents an equilibrium reaction, which shifts depending on

the concentration of the base in relation to the aniline reagent. The observed increase in $E_{11}^{\star-}$ emission for higher concentrations of base can therefore be attributed to a shift in favor of the deprotonated aniline species (see Fig. 1b). Nucleophilic attack of the SWNT would then lead to a charged SWNT intermediate that must be saturated quickly, e.g., by a proton from the solvent, yielding the defect configuration for $E_{11}^{\star-}$ emission (i.e., ortho-$L_{30}$). Similar nucleophilic reaction mechanisms have been reported for the functionalization of $C_{60}$-fullerenes via KO$^t$Bu-promoted coupling of indoles[48] and phenols[49]. In agreement with the proposed mechanism, strong $E_{11}^{\star-}$ emission (Supplementary Fig. 17 and Supplementary Table 5) was also observed for the functionalization of (6,5) SWNTs with indole, 2-iodophenol, and thiophenol, although some other emission features were introduced as well. Significant variations of the delocalization of charge can be expected for different anilines, as well as phenols and indoles, thus leading to differences in stabilization and, consequently, reactivity of the carbanion intermediate. Depending on the reactivity of the carbanion intermediate, different selectivities of $E_{11}^{\star-}$ vs. $E_{11}^{\star}$ defects may be achieved (see Supplementary Note 4). Under inert conditions, a strong increase in reactivity is observed (Supplementary Fig. 18), indicating that oxidation of the expected negatively charged intermediates (e.g., SWNT or other carbanionic intermediates, see Supplementary Note 5) and quenching of the base by moisture are inhibiting the functionalization under ambient conditions to some degree. It is also noteworthy that some nucleophilic addition reactions with nanotubes have been known for over a decade[2]; however, they usually involve very strong organic bases such as sodium hydride[50] or organolithium/organomagnesium[51,52] compounds. They have never been applied to introduce luminescent $sp^3$ defects at low concentrations and thus their selectivity for certain binding configurations is not known.

## Discussion

We have demonstrated a facile method for the functionalization of semiconducting SWNTs that selectively and exclusively creates luminescent $sp^3$ defects with a binding configuration for strongly red-shifted $E_{11}^{\star-}$ emission instead of the commonly found $E_{11}^{\star}$ emission. The reaction can be performed with various 2-haloanilines and other aniline derivatives at room temperature and in air. The final defect density is precisely and easily tuned by the concentration of the base KO$^t$Bu and the reaction time. Optimized defect densities enhance the absolute PLQYs to up to 5% for polymer-wrapped (6,5) SWNTs. The deep optical trap depth and long fluorescence lifetimes of the $E_{11}^{\star-}$ defects in (6,5) SWNTs enable their application as near-infrared single-photon emitters at room temperature with high single-photon purity. Although the defect emission wavelength of 1250 nm lies well within the second biological window and thus might be useful for in vivo imaging, emission even further in the near-infrared was achieved by functionalization of (7,5) and (10,5) nanotubes. The high selectivity of this base-mediated reaction for a specific defect-binding configuration leading to the $E_{11}^{\star-}$ emission probably relies on a nucleophilic reaction mechanism in contrast to commonly found aryl radical and reductive alkylation reactions. This type of base-mediated nucleophilic functionalization expands and complements the available chemistry for the controlled introduction of luminescent $sp^3$ defects in SWNTs and opens the path to a broader variety of functional groups and thus applications.

## Methods

**Polymer-sorting of SWNTs**. The (6,5) SWNTs were selectively dispersed from CoMoCAT raw material (Chasm SG65i-L63) by shear-force mixing (Silverson L2/Air, 10,230 r.p.m., 20 °C, 72 h) in a solution of PFO-BPy (American Dye Source,

$M_w = 40$ kg mol$^{-1}$, 0.5 g L$^{-1}$) in toluene. The (7,5) SWNTs were sorted by dispersion in toluene with PFO (American Dye Source, $M_w > 20$ kg mol$^{-1}$, 0.9 g L$^{-1}$) as the wrapping polymer. For the enrichment of (10,5) SWNTs, HiPco raw material (Unidym, Inc., Batch No. 2172) was dispersed by tip sonication with F8BT (American Dye Source, $M_W = 59$ kg mol$^{-1}$, 4 g L$^{-1}$) in toluene. Unexfoliated material was removed by centrifugation at $60,000 \times g$ (Beckman Coulter Avanti J26XP centrifuge) and filtration (polytetrafluoroethylene (PTFE) syringe filter, 5 μm pore size). To remove excess polymer, the SWNTs were collected by vacuum filtration through a PTFE membrane filter (0.1 μm pore size). Filter cakes were washed three times with toluene at 80 °C and redispersed in fresh toluene by bath sonication. Excess F8BT was removed from (10,5) SWNT dispersions by ultra-centrifugation (Optima XPN-80 centrifuge, $284,600 \times g$, 24 h). The polymer-rich supernatant was discarded and the obtained SWNT pellets were washed with THF three times before redispersion in fresh toluene.

**Introduction of $sp^3$ defects**. Polymer-wrapped SWNTs were functionalized with a range of commercially available aryl compounds (2-iodoaniline, 2-bromoaniline, 2-fluoroaniline, 5-fluoro-2-iodoaniline, 2-iodophenol, thiophenol, indole; used as received from Sigma Aldrich, ≥97%). For a step-by-step protocol, refer to Supplementary Methods 1. Briefly, an appropriate amount of aryl reagent was dissolved in toluene to achieve a final reaction concentration of 29.30 mmol L$^{-1}$. DMSO (anhydrous) and KO$^t$Bu (Sigma Aldrich, 98%) dissolved in THF (anhydrous) were added to this solution. Finally, an enriched SWNT dispersion in toluene was added to the mixture such that the SWNT concentration in the reaction mixture corresponded to an optical density of 0.3 cm$^{-1}$ at the $E_{11}$ transition. The final solvent composition was 83.3 : 8.3 : 8.3 vol-% toluene/DMSO/THF. Functionalization was performed in the dark or under illumination with UV-light (LED SOLIS-365C, Thorlabs, 365 nm, 1.9 mW mm$^{-2}$) at room temperature. The reaction was stopped after a given time by vacuum filtration of the reaction mixture through a PTFE membrane filter (0.1 μm pore size). The collected SWNTs were washed with methanol and toluene to remove unreacted reagents and side products. The resulting filter cake was redispersed by bath sonication in toluene with fresh wrapping polymer for stabilization.

**Spectroscopic characterization**. Absorption spectra were acquired with a Cary 6000i UV-VIS-NIR spectrophotometer (Varian, Inc.). PL spectra of dispersions at low excitation power and controlled temperature were recorded using a Fluorolog spectrofluorometer (HORIBA) with a 450 W xenon arc lamp and liquid-nitrogen-cooled InGaAs line camera. For acquisition of room temperature, PL spectra of dispersions, and time-correlated single-photon counting, the wavelength-tunable output of a picosecond-pulsed (~6 ps pulse width) supercontinuum laser (Fianium WhiteLase SC400) was used to excite SWNTs at the $E_{22}$ transition. PL spectra were collected with an Acton SpectraPro SP2358 spectrograph (grating blaze 1200 nm, 150 lines mm$^{-1}$) and a liquid-nitrogen-cooled InGaAs line camera (Princeton Instruments, OMA-V:1024). Wavelength-dependent PL lifetimes were measured by focusing the spectrally filtered emission onto a gated InGaAs/InP avalanche photodiode (Micro Photon Devices). The absolute PLQY was determined using an integration sphere (see Supplementary Methods 1 for details).

For low-temperature single-nanotube spectroscopy and temperature-dependent measurements, a dispersion of functionalized SWNTs was diluted with a toluene solution of polystyrene (Polymer Source, Inc., $M_w = 230$ kg mol$^{-1}$) and spin-coated onto glass slides coated with 150 nm of gold. Samples were mounted in a closed-cycle liquid helium optical cryostat (Montana Instruments Cryostation s50) and excited with a continuous wave laser diode (OBIS, Coherent, Inc., 640 nm). PL images and spectra were recorded with a thermoelectrically cooled InGaAs camera (NIRvana 640ST) coupled to an IsoPlane SCT-320 spectrograph (Princeton Instruments). A standard Hanbury–Brown and Twiss setup was employed for measurements of second-order photon-correlation function $g^{(2)}(t)$ of individual SWNTs at room temperature. The sample was excited with a supercontinuum laser (NKT Photonics, SuperK EXTREME, 6 ps pulse width, 78 MHz repetition rate, tuned to 995 nm) and the emission detected with a pair of superconducting single-photon detectors (Scontel, TCOPRS-CCR-SW-85). Detection events were recorded and correlated using a TCSPC module (PicoQuant, PicoHarp300).

**X-ray photoemission spectroscopy**. Samples were prepared by drop casting of highly functionalized (6,5) SWNT dispersions on gold-coated silicon substrates. XPS measurements were performed with a MAX 200 (Leybold–Heraeus) spectrometer equipped with an Mg Kα X-ray source (260 W; ~1.5 cm distance to the samples) and a hemispherical analyzer (EA 200; Leybold–Heraeus). Spectra were acquired in normal emission geometry with an energy resolution of ~0.9 eV. The binding energy scale was calibrated to the Au 4$f_{7/2}$ line of the substrate at 84.0 eV. Potential damage induced by X-rays was kept as low as possible. The measurements were carried out under ultra-high vacuum conditions at a base pressure of ~$3 \times 10^{-9}$ mbar.

## Data availability

The datasets generated and/or analyzed during the current study are available in the heiDATA repository: https://doi.org/10.11588/data/RI2KLV.

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

## Acknowledgements

This project has received funding from the European Research Council (ERC) under the European Union's Horizon 2020 research and innovation program (Grant agreement number 817494 "TRIFECTs"). S.Z. acknowledges funding from the Alexander von Humboldt Foundation and A.H. from the European Research Council (ERC) under the

Grant agreement number 772195 and the Deutsche Forschungsgemeinschaft (DFG, German Research Foundation) under Germany's Excellence Strategy EXC-2111-390814868.

## Author contributions

S.S., F.J.B., and S.L. performed synthesis and characterization. N.F.Z. and A.Y. contributed low-temperature spectroscopy. S.Z. and A.H. performed photon-correlation spectroscopy measurements. A.A. and M.Z. provided photoemission spectroscopy measurements. J.Z. conceived and supervised the project. S.S. and J.Z. wrote the manuscript. All authors discussed the data analysis and commented on the manuscript.

## Funding

## Competing interests

The authors declare no competing interests.
