## [Peer Review File · Nature Communications]

REVIEWER COMMENTS

Reviewer #1 (Remarks to the Author):

Settele and colleagues demonstrate a new strategy of modifying single-walled carbon nanotubes (SWCNTs) to enhance their light emission capabilities. Derivatives of aniline are combined with SWCNTs in organic solvents in the presence of potassium tert-butoxide. The results show that the execution of the process in darkness gives rise to specific binding configurations. The emergence of the E11* is clear, while the intensity of the E11* feature is greatly suppressed. A thorough analysis of the reaction conditions reveals the mechanism and the most important parameters to carry out the process effectively. Besides the basic research impact, the paper also shows considerable application potential. The authors provide convincing evidence that SWCNTs modified this way have exciting photonic characteristics. The study shows that they have appreciable single-photon purity so that they can be useful as near-infrared single-photon emitters. What is more, the paper provides another piece of evidence that functionalization of SWCNTs in organic solvents may be beneficial (10.1038/s41598-020-76716-9).

This is a remarkable contribution to the emerging field dealing with fluorescent defects in SWCNTs. The manuscript is well written, and the drawn conclusions stand on solid experimental results. Personally, I like the presented reasoning quite a lot as it is very easy to follow.

I have some questions for the authors regarding the proposed approach:

- Did the authors try different bases to elucidate the role of potassium tert-butoxide in the process more thoroughly? As rightfully reported, t-BuOK can generate radicals, which may impact the course of functionalization. It would be interesting to evaluate a base, which can exclusively support the nucleophilic pathway.
- The authors propose that there are two different functionalization mechanisms depending on whether the reaction system is illuminated with UV light or it is conducted in darkness. It is hypothesized that under these conditions, the reaction follows the radical or nucleophilic pathway, respectively. Was there any attempt to validate these suspicions by e.g. electron paramagnetic resonance? Likely, reproducing this process in an EPR spectroscope with or without UV light could provide evidence to support these claims.
- If I am not mistaken, it appears that after 180 minutes of the process (Fig. 1d), the E11*-/E11* optimum has not been reached. Similarly, Fig. 4a shows that after 240 minutes, the E11* - intensity is also highest. Did the authors examine longer modification times? It appears that if the functionalization time is extended, even higher ratios of intensities (E11*-/E11*) can be obtained. I know that according to Fig. 2d, the PLQY will suffer.

I believe that upon clarification of these minor issues, the work may be published by Nature Communications.

Kind regards,

Dawid Janas

Reviewer #2 (Remarks to the Author):

The authors report a chemical functionalization method to synthesize single-walled carbon nanotubes (SWNTs) having luminescent sp^3 -defects that show more red-shifted E11*-photoluminescence (PL) than typical E11* PL mainly observed for reported functionalized SWNTs. Previously the authors have developed chemical functionalization method of polyfluorene(PFO)-wrapped SWNTs in organic solvents using diazonium salts. In this paper, they develop a new reaction system for the PFO-wrapped SWNTs using potassium tert-butoxide (K_{Ot}Bu) and aniline derivatives under dark conditions, and succeed in preparation of the functionalized SWNTs that selectively show E11*- PL. The chemical reactions are examined by changing the amount of K_{Ot}Bu and the reaction times under UV light irradiation or dark conditions. The observed PL properties are analyzed by not only PL spectroscopies including PL quantum yield and PL lifetime measurements but also absorption and Raman spectroscopies to characterize the E11*- emission site. The functionalized SWNTs are found to be a single-photon emitter from photon antibunching. In addition, this functionalization method is applied to larger diameter SWNTs such as (7,5) and (10,5) tubes for longer wavelength PL generation. Finally, the authors propose a reaction mechanism through nucleophilic addition based on the results obtained by varying various reaction parameters. Currently, the binding configuration of the defects has been an important target to selectively control PL wavelengths of the functionalized SWNTs. The authors' findings are promising to develop further functionalized SWNTs and advanced applications such as near infrared single-photon emitters at room temperature and in-vivo bio-imaging probes. Some revision are necessary to elucidate the developed reaction system. However, after revision regarding following comments, this paper could be recommended for a publication in Nature Communications because of the novelty and significance.

1) Currently informative reviews have been published for the luminescent sp^3 -defect of the SWNTs. Therefore, the following reviews should be cited in the second sentence of the introduction part on page 3.

A. H. Brozena et al., Nat. Rev. Chem. 2019, 3, 375–392.

T. Shiraki et al., Acc. Chem. Res. 2020, 53, 1846–1859.

B. J. Gifford et al., Acc. Chem. Res. 2020, 53, 1791–1801.

2) The authors should add unnormalized PL spectra, at least for Figs. 1b and 1c. Because normalized PL spectra were used for all spectral data, which are sometimes difficult to recognize actual PL signal changes occurring in this system for readers.

3) On page 7, line 6, the authors describe “This reaction constitutes the first example of an sp³-functionalization of carbon nanotubes that exclusively leads to defects with strongly red-shifted emission (E11*–), thus exhibiting a high selectivity for this specific type of defect binding configuration”. This would be overstatement because, as the authors mentioned for ref. 17 on page 4, line 7, such nanotubes have been reported before.

4) On page 7, line 9, the authors describe “The established reaction scheme was not limited to 2-iodoaniline but was performed successfully with 2-bromoaniline and 5-fluoro-2-iodoaniline (see Supplementary Figure 2), all showing good selectivity toward E11*– defects”. However, in Supplementary Figure 2, the PL intensities of E11*– for 2-bromoaniline and 5-fluoro-2-iodoaniline functionalization are found to be much smaller than those of the 2-iodoaniline-functionalized tubes in Figure 1c based on their E11 PL. Moreover, particularly in Supplementary Figure 2b, PL shoulders are additionally observed around 1050 and 1150 nm. Therefore, the reaction efficiency and selectivity would be changed depending on the reagents. Discussion about this point is necessary to reveal the reaction systems.

5) On page 7, line 15, the authors describe “Before investigating the origin of this selectivity, the spectroscopic properties of these uniformly functionalized nanotubes and their possible application as single-photon emitters will be discussed and demonstrated”. However, to discuss about uniform functionalization, distribution of E11*– emission site on a tube needs to be investigated, for example, by hyperspectral imaging techniques for single (individual) nanotube.

6) In Fig. 2b, the absorption peak assigned to the E11*– band is rather broad compared with E11 peak, although one defect binding configuration may be formed. The origin of the peak broadness of the E11*– band needs to be discussed.

7) In Fig. 2d, PLQY of the functionalized SWNTs decreased when D/G* ratios are over 0.10 and the authors mention that it is due to being too defective, as described on page 8, line 17. On the other

hand, the PL area ratios (E_{11^*}/E_{11}) were linearly increased even such D/G^* range (> 0.10). Thus, the reason why the increase of the luminescent defects showing E_{11^*} - PL lead to the decrease in the PLQY is unclear.

8) In Fig. 3d, PL spectra of 12 individual (6,5) SWNTs functionalized with 2-iodoaniline at 4 K are shown. The observed peak positions of the E_{11^*} - PL largely changed around 1200 nm to 1300 nm. This result indicates that defect binding configuration is not limiting to only one type of luminescent defect and may have some distribution, which might be inconsistent with the claim of this paper.

9) On page 15, line, 3, the authors describe “Note that it was not possible to employ thermogravimetric analysis coupled with mass spectrometry (TGA-MS)⁴⁰ to unambiguously confirm the nature of attached functional groups due to the low defect density and limited amount of purified (6,5) SWNTs”. The authors may be able to use the functionalized SWNTs with large degree of the functionalization for the TGA-MS. In this paper, a reaction system using anionic intermediates from aniline derivatives is newly proposed. Therefore, covalently-attached molecular species on the defects need be identified. Moreover, the anionic intermediate formation in their experimental condition needs to be detected by 1H NMR or other measurements.

10) Regarding the proposed reaction mechanism, the reason why the anionic intermediates react with SWNTs not with surrounding solvents and surfactants is unclear. The authors changed the concentration of KOtBu in their experiments, but the concentration dependency of the aniline derivatives on the E_{11^*} - PL generation needs to be investigated to understand this reaction system. Relating to this point, the concentrations of the used aniline derivatives and other compounds such as indole and thiophenol should to be described in the figure captions.

11) In Supplementary Figure 16, indole, 2-iodophenol and thiophenol were used for the SWNTs functionalization. In those cases, the E_{11^*} - PL intensities seem to be small than that of the functionalized SWNTs using aniline derivatives and other peaks around 1050 nm appeared. The points need to be discussed in the manuscript.

12) On page 17, line 3 from the bottom, an oxygen effect on PL is described. If the oxidation occur for the negatively charged SWNT intermediate, the reaction should also form sp^3 defect, which could contribute the luminescent defect formation. Therefore, other factors of the oxygen need to be considered to explain the increase of the E_{11^*} - intensity in the inert condition.

13) In Supplementary Figure 7a, a strange PL peak around 1050 nm appeared only at low temperature such as 4 and 160 K, which is not observed for the solution samples and other

functionalized SWNTs in Supplementary Figure 8a. The possible origin of this peak should be suggested.

14) When comparing Supplementary Figures 8a and 10a, the intensities of the E11*- PL peaks are remarkably different between the film and the solution states, in which the intensities of E11 and E11* are similarly observed. In sharp contrast, Supplementary Figures 7a and 9a show comparable intensities of the E11*- PL peaks even in the film and the solution states. The difference needs to be discussed to understand the conditions of each sample.

Reviewer #3 (Remarks to the Author):

Settele et al. reports on a method for the selective creation of sp³ defects in single-walled carbon nanotubes. Although defects giving rise to similar spectral features have been reported in several other works, the detailed discussion of the reaction mechanism and the selectivity developed here is of interest to the carbon nanotube community. This study does help expand the toolbox for controlled creation of defects. The work is quite thorough and the manuscript is well written. I have a few comments/concerns, though, that the authors should address before I would recommend the publication of the manuscript.

1. My biggest concern is the nucleophilic reaction mechanism proposed here. The origin of the E11*- peak has remained inconclusive. They were first observed in oxygen-doped CNTs (see e.g. ACS Nano 2014, 8, 10782) and then in diazonium- (e.g. Nat Photon 2017, 11, 577) and other (JACS 2017, 139, 4859) molecularly-functionalized CNTs. While they have been mostly assigned to specific configurations of the sp³ carbons, there has been compelling evidence indicating the charge-negative nature of the E11*- defects (ACS Cent. Sci. 2019, 5, 1786), which is in contradiction with the nucleophilic reaction mechanism proposed by the authors. The authors should address this issue by performing control experiments that would determine the nature of the quasiparticles associated with the E11*- defects.

2. From the perspective of utilizing these defects as single photon sources, narrowing down the emission spectral range from 1000 – 1300 nm to 1200 – 1300 nm doesn't really pose a significant advantage. The challenge is more of the homogeneous broadening rather than the lack of abundance of the E11*- defects. The authors may want to reorganize some of the statements in this regard.

3. The definition of the separate QYs for the E11*- and E11 is inappropriate. Excitons generated in the E11 state could have two routes to give out photons: either emit in the pristine segments or diffuse to the defects and emit at the E11*- energy. By the definition of QY, for certain absorbance, it is only meaningful to count the total emission from both the E11 and E11*- states. The authors should adjust the discussion of the E11 and E11*- peaks in Fig. 2d to PL intensity rather than separate QYs.

4. page 12: the signatures in the E11 region has been solely assigned to localization of excitons. Previous studies have indicated that these are associated with certain sp³ configurations (e.g. ACS Nano 2017, 11, 10785). The authors should consider expanding their discussion on this.

Point-by-point Response

Manuscript # NCOMMS-20-46170-T

We thank all reviewers for their careful consideration of this manuscript. Changes that were applied to the revised manuscript or Supplementary Information are highlighted in **bold print** in the following point-by-point response. The revised versions of the manuscript and Supplementary Information with marked changes are provided separately for review.

Reviewer #1 (Remarks to the Author):

Settele and colleagues demonstrate a new strategy of modifying single-walled carbon nanotubes (SWCNTs) to enhance their light emission capabilities. Derivatives of aniline are combined with SWCNTs in organic solvents in the presence of potassium tert-butoxide. The results show that the execution of the process in darkness gives rise to specific binding configurations. The emergence of the E₁₁^{*-} is clear, while the intensity of the E₁₁^{*} feature is greatly suppressed. A thorough analysis of the reaction conditions reveals the mechanism and the most important parameters to carry out the process effectively. Besides the basic research impact, the paper also shows considerable application potential. The authors provide convincing evidence that SWCNTs modified this way have exciting photonic characteristics. The study shows that they have appreciable single-photon purity so that they can be useful as near-infrared single-photon emitters. What is more, the paper provides another piece of evidence that functionalization of SWCNTs in organic solvents may be beneficial (10.1038/s41598-020-76716-9).

This is a remarkable contribution to the emerging field dealing with fluorescent defects in SWCNTs. The manuscript is well written, and the drawn conclusions stand on solid experimental results. Personally, I like the presented reasoning quite a lot as it is very easy to follow.

I have some questions for the authors regarding the proposed approach:

- Did the authors try different bases to elucidate the role of potassium tert-butoxide in the process more thoroughly? As rightfully reported, t-BuOK can generate radicals, which may impact the course of functionalization. It would be interesting to evaluate a base, which can exclusively support the nucleophilic pathway.

RESPONSE: The use of different bases might indeed open up new ways to tune the selectivity of the reaction. However, as argued below, a suitable base/arene system must meet a significant number of criteria. Apart from KO^tBu no such base could be identified within the scope of this work.

The highest selectivity for E₁₁^{*-} defects was achieved through the use of aniline derivatives, however, anilines are very poor acids with a pK_a value of 28.7 (for 2-fluoroaniline in DMSO, *J.*

Am. Chem. Soc. **1988**, *110*, 2964). A very strong base is therefore necessary to achieve efficient deprotonation and thus generation of reactive carbanion intermediates. Related studies by Li *et al.* on the functionalization of C₆₀-fullerenes with indole found a significant decrease in functionalization for weaker base systems compared to KO^tBu/DMSO (see *J. Org. Chem.* **2015**, *80*, 10605–10610). Considering the lower pK_a of indole (pK_a 21.0) compared to aniline and that the reactivity of fullerenes is in general higher compared to SWNTs, we did not test weaker base systems.

The opposite issue arises when considering very strong bases such as butyl lithium. Organolithium compounds were found to lead to significant side-wall functionalization of SWNTs (see *Chem. Rev.* **2006**, *106*, 1105–1136, *J. Am. Chem. Soc.* **2012**, *134*, 18101–18108, *J. Am. Chem. Soc.* **2006**, *128*, 6683–6689) and hence are not suitable for the controlled introduction of a few luminescent sp³-defects. Lastly, the employed bases should not act as reducing agents as this could lead to a Billups-Birch type functionalization (*Nano Letters* **2004**, *4*, 1257–1260).

Overall a strong base is necessary for deprotonation of the aniline derivatives. While many bases might be suitable for deprotonation, they cannot be used as they also show undesired side reactions with SWNTs. The stringent criteria for a suitable base system are fulfilled by KO^tBu in DMSO. The pK_a of KO^tBu/DMSO (32.2) is sufficient for efficient deprotonation of anilines and side-reactions can be easily suppressed.

In contrast to anilines, phenols represent a significantly stronger acid in DMSO (pK_a 18.0, *J. Org. Chem.* **1984**, *49*, 1424–1427) and can be deprotonated by weaker bases such as potassium carbonate. We performed a test reaction with 2-iodophenol and potassium carbonate under UV-light irradiation to enhance the functionalization process, however, it only resulted in an extremely weak additional sideband at 1250 nm (corresponding to E₁₁*⁻ emission, see Figure).

Finally, we agree with the reviewer on the importance of the base and hence expanded our discussion about this topic in the **revised Supplementary Information under “Mechanistic Considerations”**.

Functionalization of PFO-BPy wrapped (6,5)-SWNTs with 2-iodophenol (29.30 mmol L⁻¹) under irradiation of UV-light (365 nm) for 60 minutes in the presence of 2 equivalents of **potassium carbonate** (58.60 mmol L⁻¹).

- The authors propose that there are two different functionalization mechanisms depending on whether the reaction system is illuminated with UV light or it is conducted in darkness. It is hypothesized that under these conditions, the reaction follows the radical or nucleophilic pathway, respectively. Was there any attempt to validate these suspicions by e.g. electron paramagnetic resonance? Likely, reproducing this process in an EPR spectroscope with or without UV light could provide evidence to support these claims.

RESPONSE: While we considered electron paramagnetic resonance (EPR) spectroscopy, we did not perform such measurements as the formation of additional radical species is expected. These other radical species, which do not engage in the functionalization process, would make analysis of the EPR data impossible. EPR responses can therefore not be used as evidence or counter argument of our proposed mechanism. For example, anilines are expected to form amine radical intermediates upon deprotonation and single electron transfer as reported by Ghosh *et. al.* (*Org. Lett.* **2019**, *21*, 6690–6694) and shown by EPR measurements. Such amine radicals are not expected to lead to covalent functionalization of SWNTs and can be ruled out as the source of E_{11}^{*-} emission as this feature is also observed for functionalization with 2-iodophenol, thiophenol and indole.

- If I am not mistaken, it appears that after 180 minutes of the process (Fig. 1d), the E_{11}^{*-}/E_{11}^{*} optimum has not been reached. Similarly, Fig. 4a shows that after 240 minutes, the E_{11}^{*-} intensity is also highest. Did the authors examine longer modification times? It appears that if the functionalization time is extended, even higher ratios of intensities (E_{11}^{*-}/E_{11}^{*}) can be obtained. I know that according to Fig. 2d, the PLQY will suffer.

RESPONSE: Longer reaction times are expected to lead to higher ratios of E_{11}^{*-} / E_{11}^{*} emission intensities. However, not only will the PLQY suffer (as shown in Figure 2d) but also the yield of the redispersion step due to progressing aggregation of SWNTs during the functionalization. Hence, the combination of decreased yield and significantly lower PLQY represents a significant problem for the investigation of longer reaction times. At this point we would like to highlight that this decrease in yield is not directly connected to the reaction time but rather the defect density. For (7,5) SWNTs, as presented in Fig. 4a, the defect density is still low and longer reactions times are a simple way to achieve higher E_{11}^{*-} / E_{11}^{*} intensity ratios.

We acknowledge that this aggregation effect at the highest defect densities is of importance for the reader and added a comment in the **revised Supplementary Information under the work-up section of the „Step by Step Reaction Protocol“**.

I believe that upon clarification of these minor issues, the work may be published by Nature Communications.

RESPONSE: We thank the reviewer for their comments and positive assessment.

Reviewer #2 (Remarks to the Author):

The authors report a chemical functionalization method to synthesize single-walled carbon nanotubes (SWNTs) having luminescent sp³-defects that show more red-shifted E11*-photoluminescence (PL) than typical E11* PL mainly observed for reported functionalized SWNTs. Previously the authors have developed chemical functionalization method of polyfluorene(PFO)-wrapped SWNTs in organic solvents using diazonium salts. In this paper, they develop a new reaction system for the PFO-wrapped SWNTs using potassium tert-butoxide (KOtBu) and aniline derivatives under dark conditions, and succeed in preparation of the functionalized SWNTs that selectively show E11*- PL. The chemical reactions are examined by changing the amount of KOtBu and the reaction times under UV light irradiation or dark conditions. The observed PL properties are analyzed by not only PL spectroscopies including PL quantum yield and PL lifetime measurements but also absorption and Raman spectroscopies to characterize the E11*- emission site. The functionalized SWNTs are found to be a single-photon emitter from photon antibunching. In addition, this functionalization method is applied to larger diameter SWNTs such as (7,5) and (10,5) tubes for longer wavelength PL generation. Finally, the authors propose a reaction mechanism through nucleophilic addition based on the results obtained by varying various reaction parameters. Currently, the binding configuration of the defects has been an important target to selectively control PL wavelengths of the functionalized SWNTs. The authors' findings are promising to develop further functionalized SWNTs and advanced applications such as near infrared single-photon emitters at room temperature and in-vivo bio-imaging probes. Some revision are necessary to elucidate the developed reaction system. However, after revision regarding following comments, this paper could be recommended for a publication in Nature Communications because of the novelty and significance.

1) Currently informative reviews have been published for the luminescent sp³-defect of the SWNTs. Therefore, the following reviews should be cited in the second sentence of the introduction part on page 3.

A. H. Brozena et al., Nat. Rev. Chem. 2019, 3, 375–392.

T. Shiraki et al., Acc. Chem. Res. 2020, 53, 1846–1859.

B. J. Gifford et al., Acc. Chem. Res. 2020, 53, 1791–1801.

RESPONSE: We thank the reviewer for pointing out these recent overview articles. **All of them are now cited (Refs. 7-9) in the second sentence of the introduction.**

2) The authors should add unnormalized PL spectra, at least for Figs. 1b and 1c. Because normalized PL spectra were used for all spectral data, which are sometimes difficult to recognize actual PL signal changes occurring in this system for readers.

RESPONSE: There are several reasons why we only show normalized PL spectra for the defect density series:

- (1) To stop the reaction and remove side products, our protocol includes a filtration, washing and redispersion sequence (see Experimental Section). Note, it is not possible to simply record the PL spectra during the reaction in-situ. As with pristine SWNTs, the yield of the redispersion step strongly varies and depends on parameters such as sonication power, environmental humidity and temperature. Hence, the resulting dispersions have different concentrations of the dispersed SWNTs and consequently different absolute PL intensities.
- (2) While the absorption spectrum of the SWNTs is unaffected by low-level functionalization, the samples with higher defect densities display a significant bleaching of the main absorption bands. Hence, the effective absorption cross-section for the functionalized SWNTs also depends on defect density preventing a correction for the yield of the redispersion step.
- (3) The PL measurements were performed by focusing the excitation laser into a cuvette through an objective. This configuration has the advantage that the near-infrared absorption of toluene does not affect the PL spectra due to the extremely short path length within the liquid. However, even slight differences in the quality of the focus have an impact on the absolute PL intensities, while the spectrum is usually unaffected by minor changes in focus.

As a result, we think that a comparison of the absolute PL intensities across these sample series is unreliable and should be avoided. These considerations underscore the importance of **absolute PLQY measurements** as performed using an integrating sphere (see **Figure 2d**), which give a direct measure of the emission efficiencies.

3) On page 7, line 6, the authors describe “This reaction constitutes the first example of an sp³-functionalization of carbon nanotubes that exclusively leads to defects with strongly red-shifted emission (E₁₁*⁻), thus exhibiting a high selectivity for this specific type of defect binding configuration”. This would be overstatement because, as the authors mentioned for ref. 17 on page 4, line 7, such nanotubes have been reported before.

RESPONSE: The reviewer is correct in pointing out that Saha *et al.* (*Nat. Chem.* **2018**, *10*, 1089-1095) showed that functionalization of achiral (11,0) SWNTs leads to strongly red-shifted emission (E₁₁*⁻) with high selectivity. This selectivity originates from the equivalence of binding configurations due to the high symmetry of “zigzag” nanotubes. However, this high symmetry effect only applies to zigzag SWNTs. In contrast to that, our method enables the selective functionalization of chiral SWNTs (here (6,5), (7,5) and (10,5) nanotubes) that otherwise show multiple PL features due to various possible binding configurations. To emphasize this aspect, we rephrased our statement to “**chiral** carbon nanotubes”.

4) On page 7, line 9, the authors describe “The established reaction scheme was not limited to 2-iodoaniline but was performed successfully with 2-bromoaniline and 5-fluoro-2-iodoaniline (see Supplementary Figure 2), all showing good selectivity toward E_{11}^{*-} defects”. However, in Supplementary Figure 2, the PL intensities of E_{11}^{*-} for 2-bromoaniline and 5-fluoro-2-iodoaniline functionalization are found to be much smaller than those of the 2-iodoaniline-functionalized tubes in Figure 1c based on their E_{11} PL. Moreover, particularly in Supplementary Figure 2b, PL shoulders are additionally observed around 1050 and 1150 nm. Therefore, the reaction efficiency and selectivity would be changed depending on the reagents. Discussion about this point is necessary to reveal the reaction systems.

RESPONSE: This topic is briefly discussed on pages 15/16 of the main manuscript.

We assume that the selectivity of the functionalization process relies on the resonance structure of the reactive intermediate. As this reactive intermediate is different for each reagent, changes in reactivity and selectivity are expected for 2-bromoaniline and 5-fluoro-2-iodoaniline. In particular the steric interactions of 5-fluoro-2-iodoaniline can potentially promote different reactions paths.

We have addressed this issue in more detail in our response to comment 11 (see below).

We acknowledge that the change in reactivity and selectivity is of significance for the reader and **expanded the corresponding discussion in the revised manuscript (page 7)**. We also **added the respective reaction times** for the functionalizations performed with 2-bromoaniline and 5-fluoro-2-iodoaniline (**caption Supplementary Figure 3**).

The origin of emission shoulders at 1050 nm is discussed in detail in our response to comment 13 (see below).

5) On page 7, line 15, the authors describe “Before investigating the origin of this selectivity, the spectroscopic properties of these uniformly functionalized nanotubes and their possible application as single-photon emitters will be discussed and demonstrated”. However, to discuss about uniform functionalization, distribution of E_{11}^{*-} emission site on a tube needs to be investigated, for example, by hyperspectral imaging techniques for single (individual) nanotube.

RESPONSE: We agree with the reviewer that additional experiments would be necessary to corroborate a uniform spatial distribution of E_{11}^{*-} defects on individual SWNTs. However, we did not intend to imply such a property and the term “uniformly” was used to highlight the selectivity of E_{11}^{*-} defect creation rather than the distribution of different types of defects (incl. E_{11}^{*}) among individual nanotubes in an ensemble. We **deleted the term “uniformly” on page 8** to avoid further confusion and thank the reviewer for pointing this out.

6) In Fig. 2b, the absorption peak assigned to the E_{11}^{*-} band is rather broad compared with E_{11} peak, although one defect binding configuration may be formed. The origin of the peak broadness of the E_{11}^{*-} band needs to be discussed.

RESPONSE: At this point, we can only speculate about the origin of the absorption peak broadness. Furthermore, it is difficult to evaluate whether this peak width is unusual because there are only very few published absorption spectra showing a clear defect band. Samples with increased broadness of absorption band show very high D/G^+ ratios (>0.2), which can also lead to stronger aggregation. This leads us to assume that the broadening occurs due to an increased scattering background caused by aggregation of SWNTs in dispersion. **We have addressed the broadness of the absorption peak in the revised manuscript (see Figure caption for Fig. 2b).**

7) In Fig. 2d, PLQY of the functionalized SWNTs decreased when D/G^+ ratios are over 0.10 and the authors mention that it is due to being too defective, as described on page 8, line 17. On the other hand, the PL area ratios (E_{11}^{*-}/E_{11}) were linearly increased even such D/G^+ range (>0.10). Thus, the reason why the increase of the luminescent defects showing E_{11}^{*-} - PL lead to the decrease in the PLQY is unclear.

RESPONSE: The increase in PL area ratios (E_{11}^{*-}/E_{11}) does not represent a contradiction to a decreasing PLQY for D/G^+ ratios higher than 0.1. With increasing number of defects E_{11} excitons become more likely to be trapped at E_{11}^{*-} defect sites, thus the PL area ratio (E_{11}^{*-}/E_{11}) continuously grows for higher D/G^+ ratios independent of the total PLQY. However, above a D/G^+ ratio of 0.1, the introduction of more and more sp^3 -carbons disrupts the sp^2 lattice creating actual quenching sites for mobile excitons and thus affect both E_{11}^{*-} and E_{11} emission. Such behaviour was previously observed for luminescent sp^3 -defects introduced by diazonium chemistry (Piao *et al. Nat. Chem.* **2013**, *5*, 840-845) in water as well as in organic solvent (Berger *et al. ACS Nano* **2019**, *13*, 9259–9269) and is in agreement with the negligible PL observed for highly functionalized SWNTs (see *J. Am. Chem. Soc.* **2008**, *130*, 6795–6800). To further support this, we plotted the D/G^+ ratios vs. the PL area ratios (E_{11}^{*-}/E_{11}) corresponding to the data in Fig. 2d. A linear correlation is observed similar to Fig. S4b. We have **clarified our statement in the revised manuscript on page 9 (first paragraph).**

Integrated E_{11}^{*-}/E_{11} emission ratios vs. integrated Raman D/G^+ ratios and linear fit as metric of defect density for data in Figure 2d. Blue triangle: data point for pristine (6,5) SWNTs.

8) In Fig. 3d, PL spectra of 12 individual (6,5) SWNTs functionalized with 2-iodoaniline at 4 K are shown. The observed peak positions of the E_{11}^{*-} PL largely changed around 1200 nm to 1300 nm. This result indicates that defect binding configuration is not limiting to only one type of luminescent defect and may have some distribution, which might be inconsistent with the claim of this paper.

RESPONSE: The observed changes in peak position of E_{11}^{*-} PL between 1200 to 1300 nm are typical distributions due to variations in the dielectric environment as reported in previous studies by Saha *et al.* (*Nat. Chem.* **2018**, *10*, 1089-1095) and He *et al.* (*Nat. Photonics* **2017**, *11*, 577–582), who also interpreted these signals as arising from a single binding configuration in a heterogeneous environment. The boundary between E_{11}^* and E_{11}^{*-} defects can be drawn around 1200 nm. This assignment is also supported by calculations for different defect binding configurations for (6,5) by Saha *et al.* (*Nat. Chem.* **2018**, *10*, 1089). We have clarified this in the **revised manuscript on page 13**.

9) On page 15, line, 3, the authors describe “Note that it was not possible to employ thermogravimetric analysis coupled with mass spectrometry (TGA-MS)⁴⁰ to unambiguously confirm the nature of attached functional groups due to the low defect density and limited amount of purified (6,5) SWNTs”. The authors may be able to use the functionalized SWNTs with large degree of the functionalization for the TGA-MS. In this paper, a reaction system using anionic intermediates from aniline derivatives is newly proposed. Therefore, covalently-attached molecular species on the defects need be identified. Moreover, the anionic intermediate formation in their experimental condition needs to be detected by ¹H NMR or other measurements.

RESPONSE: The defect concentration for which functionalized (6,5) SWNTs are expected to show significant E_{11}^{*-} emission is in the range of nmol L⁻¹, which is below the detection limit of TGA-MS. For an extremely high degree of functionalization TGA-MS might be possible, but it is unlikely that the defects at such high defect densities still reflect the type of defects (binding configuration) at low functionalization levels. Hence, the data would not provide an unambiguous answer. For this reason, we performed X-ray photoelectron spectroscopy (XPS) to confirm the nature of the attached functional groups. The XPS measurements (Figure 5 and Figure S16) could show an unambiguous signature corresponding to the F1s core level. Thus, the covalently attached molecule can be clearly identified in Figure 5 as 2-fluoroaniline.

Regarding the formation of the anionic intermediate: The deprotonation of aniline derivatives by KO^tBu in DMSO has been frequently reported in literature (*e.g.* *Eur. J. Org. Chem.* **2018**, 3454–3463, *Org. Lett.* **2019**, *21*, 6690-6694, *Eur. J. Org. Chem.* **2019**, 4538–4545). In general, the pK_a of anilines is around 29 (28.7 for 2-fluoroaniline in DMSO, *J. Am. Chem. Soc.* **1988**, *110*, 2964) and should thus be easily deprotonated by KO^tBu (pK_a 32.2 in DMSO, *J. Org. Chem.* **1980**, *45*, 3295-3299). The subsequent delocalization of the negative charge under formation of carbanions is a well-established textbook equilibrium (*e.g.* Organic Chemistry by Clayden, Greeves & Warren; Springer, 2nd Edition, p.195).

However, we understand that the mechanism initially proposed in Fig. 6 might be misleading in this regard. It is not clear which of the possible anionic intermediates is participating in the

functionalization step. Hence, we **rearranged the presentation of the mechanism in the revised Fig. 6**. It now shows the equilibrium between the anionic intermediates. We also **added a reference for the deprotonation step of the aniline derivative (reference No. 43, *Eur. J. Org. Chem.* 2018, 3454–3463)**

10) Regarding the proposed reaction mechanism, the reason why the anionic intermediates react with SWNTs not with surrounding solvents and surfactants is unclear. The authors changed the concentration of KOtBu in their experiments, but the concentration dependency of the aniline derivatives on the E₁₁*- PL generation needs to be investigated to understand this reaction system. Relating to this point, the concentrations of the used aniline derivatives and other compounds such as indole and thiophenol should to be described in the figure captions.

RESPONSE: Similar to many previously reported procedures (*Sci Rep* 2020, 10, 19877; *ACS Nano* 2020, 14, 715–723) functionalization in this work was performed with a large excess of the reactant compared to the SWNTs. To investigate the concentration dependence of the aniline we additionally functionalized (6,5) SWNTs following the reported procedure with different concentrations of 2-iodoaniline, while the ratio of 2-iodoaniline to KOtBu was constant at 1:2. Resulting PL spectra show that the concentrations of 2-iodoaniline used in this work was indeed in the saturation regime and increasing/decreasing the concentration does not significantly alter the functionalization process. For much lower concentrations the E₁₁*⁻ emission feature drops significantly. This is expected, because (i) the amount of reactive intermediate is greatly reduced, and (ii) the relative quenching of KOtBu by moisture increases as the reactions are performed in an open flask.

To further understand the effect of the concentration of 2-iodoaniline, we conducted functionalizations with various concentrations of 2-iodoaniline, while the concentration of KOtBu was kept constant. These conditions led to an increase of the ratio of 2-iodoaniline to KOtBu. As functionalization is still performed with a large excess of the reactant compared to SWNTs the E₁₁*⁻ emission is higher for higher 2-iodoaniline/KOtBu ratios in agreement with Figure 1c.

We added the results of both experiments to the Supplementary Information (**new Supplementary Figure 2**) and included a corresponding sentence in the main text (**on page 6**) of the revised manuscript.

We thank the reviewer for pointing out the missing concentrations of used aniline derivatives and other compounds. **We added these values to the corresponding Figure captions.**

New Figure S2: PL spectra of (6,5) SWNTs functionalized with varying concentrations of 2-iodoaniline and 2 eq. of KOtBu (a) and varying concentrations of 2-iodoaniline and fixed concentration of KOtBu (58.60 mmol L⁻¹) (b). Functionalization was performed in the dark for 10 minutes.

11) In Supplementary Figure 16, indole, 2-iodophenol and thiophenol were used for the SWNTs functionalization. In those cases, the E₁₁*⁻ PL intensities seem to be small than that of the functionalized SWNTs using aniline derivatives and other peaks around 1050 nm appeared. The points need to be discussed in the manuscript.

RESPONSE: We agree with the reviewer and **greatly expanded our discussion on this topic in the revised manuscript (pages 18/19) and Supplementary Information (Fig. S17 Functionalization with Non-Aniline Reagents).**

Anilines, indoles and phenols represent completely different substance classes and thus significant changes in reactivity and selectivity can be expected due to different structures of the carbanion intermediate. Stabilization and position of the reactive centre of the carbanion can differ significantly. For example, Li *et al.* showed that the reaction of C₆₀-fullerenes with indole in the presence of KOtBu/DMSO occurs *via* functionalization in C₃-position of the indole (*Org. Lett.* **2017**, *19*, 1192-1195). In contrast to that, phenol attacks in the C₄-position (*J. Org. Chem.* **2018**, *83*, 5431-5437). In case of anilines the lowest relative energy of the carbanion is expected in C₂-position (*J. Struct. Chem.* **2010**, *22*, 345-356), however, steric interaction could limit an attack in this position.

Overall the lowest energy path through the potential energy surface of the reaction varies between different substance classes and can lead to significant changes in reactivity for E₁₁*⁻ defects. Thus, when the introduction of E₁₁*⁻ is hindered, other functionalization processes (*e.g.* radical functionalization) might become more kinetically favoured. This can result in additional shoulders and sidebands in the PL spectrum and overall lower selectivity for one specific defect emission. This concept is supported by the very similar PL spectra obtained for functionalization with 2-iodophenol and thiophenol, as they represent similar substance classes and are expected to follow similar reaction paths. Finally, in case of anilines it was found that the introduction of E₁₁*⁻ defects

can be controlled through the deprotonation equilibrium and thus KO^tBu concentration (see Figure 1b). This equilibrium is different for each substance class as their pK_a values change.

The origin of the shoulder at 1050 nm is discussed in detail in our response to comment 13.

12) On page 17, line 3 from the bottom, an oxygen effect on PL is described. If the oxidation occur for the negatively charged SWNT intermediate, the reaction should also form sp³ defect, which could contribute the luminescent defect formation. Therefore, other factors of the oxygen need to be considered to explain the increase of the E11*⁻ intensity in the inert condition.

RESPONSE: Oxidation of negatively charged SWNT intermediates does not have to create sp³ defects. Li *et al.* (*Org. Lett.* **2017**, *19*, 1192-1195), who studied the functionalization of C₆₀-fullerenes with indoles using KO^tBu/DMSO in much detail, observed no reaction at all when performing the reaction in air. Such behaviour indicates that the carbon double bond is fully regenerated after oxidation of the charged intermediate and dissociation of the functional group. Such an oxidation mechanism is likely to also occur in the case of SWNTs.

However, we agree that other effects should be considered. For example, the moisture level in the reaction mixture is also reduced under inert conditions. We added this aspect to our discussion **on page 19 of the revised manuscript.**

13) In Supplementary Figure 7a, a strange PL peak around 1050 nm appeared only at low temperature such as 4 and 160 K, which is not observed for the solution samples and other functionalized SWNTs in Supplementary Figure 8a. The possible origin of this peak should be suggested.

RESPONSE: Emission features around 1050 nm are not exclusively observed for functionalized SWNTs but also appear for pristine SWNTs as reported by Kadria-Vili *et al.* (*J. Phys. Chem. C* **2016**, *120*, 23898–23904) and are commonly labelled as the Y₁ band. It is assumed that this feature originates from defects that are unintentionally introduced during the growth or processing (dispersion) of SWNTs. The observed Y₁ PL intensity already varies for untreated (“pristine”) SWNTs. Furthermore, the functionalization of SWNTs represents an additional processing step with varying reaction times, sonication and annealing steps (for film preparation as presented in Supplementary Figure 8a) and can lead to batch-to-batch variations. Indeed, we found several individual SWNTs in our untreated (“pristine”) reference sample that showed PL signals around 1050 nm, as illustrated in the Figure below. Therefore, the fact that the Y₁ signal appears in Fig. S8a, but not in Fig. S9a (in the revised manuscript), is likely due to batch-to-batch variation. Moreover, the Y₁ signal is barely observed in the spectra of dispersions, because these measurements were performed at room temperature and the concomitant increase in E₁₁ linewidth covers the Y₁ shoulder. As evident from Fig. S8a, the Y₁ band becomes almost unresolvable at temperatures around 300 K. **We added a corresponding discussion to Supplementary Figure 8 (previously S7).**

PL spectra of untreated (“pristine”) (6,5) SWNTs embedded in a polystyrene matrix recorded at 4 K.

14) When comparing Supplementary Figures 8a and 10a, the intensities of the E_{11}^{*-} PL peaks are remarkably different between the film and the solution states, in which the intensities of E11 and E_{11}^* are similarly observed. In sharp contrast, Supplementary Figures 7a and 9a show comparable intensities of the E_{11}^{*-} PL peaks even in the film and the solution states. The difference needs to be discussed to understand the conditions of each sample.

RESPONSE: The measurements for Supplementary Figures 9a and 11a (of the revised manuscript) were conducted on the same sample, while a different sample had to be used for Supplementary Figures 8a and 10a (of the revised manuscript). As seen in Supplementary Figure 8a the temperature-dependent change in defect emission is significantly smaller for E_{11}^{*-} defects compared E_{11}^* defects. Thus, we chose to perform measurements in films of highly functionalized SWNTs to ensure sufficient signal from E_{11}^{*-} defects. However, temperature-dependent PL measurements in dispersions gave the clearest results at low defect concentrations because it reduced aggregation during the course of the lengthy measurement. Hence, we chose to use two different functionalization batches for measurements in film and dispersion when investigating E_{11}^{*-} defect emission. To avoid any confusion or errant assumptions by the readers we clarified the use of different samples/batches in the figure captions of the **revised Supplementary Information Figures 10 and 11.**

Reviewer #3 (Remarks to the Author):

Settele et al. reports on a method for the selective creation of sp^3 defects in single-walled carbon nanotubes. Although defects giving rise to similar spectral features have been reported in several other works, the detailed discussion of the reaction mechanism and the selectivity developed here is of interest to the carbon nanotube community. This study does help expand the toolbox for controlled creation of defects. The work is quite thorough and the manuscript is well written. I have a few comments/concerns, though, that the authors should address before I would recommend the publication of the manuscript.

RESPONSE: We thank the reviewer for the positive assessment and have addressed all comments/concerns in detail below.

1. My biggest concern is the nucleophilic reaction mechanism proposed here. The origin of the E_{11}^{*-} peak has remained inconclusive. They were first observed in oxygen-doped CNTs (see e.g. ACS Nano 2014, 8, 10782) and then in diazonium- (e.g. Nat Photon 2017, 11, 577) and other (JACS 2017, 139, 4859) molecularly-functionalized CNTs. While they have been mostly assigned to specific configurations of the sp^3 carbons, there has been compelling evidence indicating the charge-negative nature of the E_{11}^{*-} defects (ACS Cent. Sci. 2019, 5, 1786), which is in contradiction with the nucleophilic reaction mechanism proposed by the authors. The authors should address this issue by performing control experiments that would determine the nature of the quasiparticles associated with the E_{11}^{*-} defects.

RESPONSE: The question whether the new defect peak might possibly be related to a charged defect can be answered easily and unambiguously by in-situ photoluminescence spectra of electrolyte-gated nanotubes with the E_{11}^{*-} defects. We have done this experiment and provide the data here for the review process:

Electrolyte-gating is a technique where the carbon nanotubes (here a dropcast network of (6,5) SWNTs with a moderate density of E_{11}^{*-} defects) together with source/drain electrodes represent the working electrode of an electrochemical cell. The gate electrode (large gold pad) is the counter electrode (a reference electrode can be included if required). The electrolyte in our case was an ionic gel based on the ionic liquid [EMIM][FAP] which has a large electrochemical window (see Figure). This is a well-established technique both for nanotubes but also organic semiconductors. The samples were encapsulated under nitrogen to enable both electron and hole accumulation.

In-situ PL spectroscopy of electrolyte-gated carbon nanotubes enables independent determination of the charge carrier density, as the current between the source and drain electrode (channel length 20 μm) is monitored (at a very low drain voltage) concurrently. For nanotubes without sp^3 -defects PL quenching of the E_{11} emission (possibly Auger quenching, see *Nano Lett.* **2009**, 9, 3477-3481) and the emergence of trion emission is typically observed with both hole and electron accumulation (regarding trion emission for (6,5) SWNTs without defects refer to *ACS Nano* **2014**, 8, 8477-8486 & *ACS Nano* **2020**, 14, 2412-2423). The maximum E_{11} emission is found when the network is not doped at all, which is confirmed by a very low (leakage level) drain current (minimum carrier concentration).

The transfer characteristic shown in the figure (top left) confirm hole and electron accumulation for negative and positive gate voltages, respectively. In a gate voltage range from 0 V to +1 V the network is not charged (no mobile carriers). The gap between the onset of hole and electron transport is directly related to the bandgap of the nanotubes and their redox potentials (see *Angew. Chem., Int. Ed.* 2009, 48, 7655). The offset from 0 V is typical and caused by residual water and oxygen. The superimposed normalized intensity of the E_{11} emission, which drops steeply at around -0.2 V (hole doping) and +1.4 V (electron doping), confirms the different ranges of charge accumulation and doping. The normalized E_{11}^{*-} emission intensity follows this trend precisely. The corresponding PL spectra (absolute intensities and normalized to the E_{11} peak for electron accumulation and holes accumulation) show unambiguously that the E_{11}^{*-} emission is actually quenched by doping. It is even more strongly quenched than the mobile exciton E_{11} emission. This behavior is expected and also found for E_{11}^* defects (not shown here). At very high doping levels, both E_{11} and E_{11}^{*-} are strongly quenched and the weak trion emission (originating from E_{11} excitons) becomes dominant (this has been shown previously, see *ACS Nano* 2014, 8, 8477-8486 & *ACS Nano* 2020, 14, 2412-2423).

It is very clear that the maximum E_{11}^{*-} emission with its peak at 1252 nm is found in the neutral state of the functionalized (6,5) SWNTs and hence the **emission is due to a different binding configuration** (as usually assumed based on the calculations by the Tretiak group) and **not** a trion emission of a negatively charged E_{11}^* defect. **Consequently, there is no contradiction to the proposed mechanism.**

2. From the perspective of utilizing these defects as single photon sources, narrowing down the emission spectral range from 1000 – 1300 nm to 1200 – 1300 nm doesn't really pose a significant advantage. The challenge is more of the homogeneous broadening rather than the lack of abundance of the E11*- defects. The authors may want to reorganize some of the statements in this regard.

RESPONSE: The inhomogeneous distribution of quantum emitter wavelengths might initially appear as a minor factor in the development of nanotube quantum defects as single photon sources. The reviewer has correctly pointed out that the homogeneous emission line broadening is a major limitation to photon indistinguishability and thus one of the main issues for applications. In this respect, cavity-enhancement of the radiative emission rate *via* the Purcell effect has been shown recently to improve the photon indistinguishability by bringing the homogeneous linewidth closer to the Fourier-transform limit (as demonstrated with plasmonic cavities by Luo *et al. Nano Lett.* **2019**, *19*, 9037–9044). One should note in this context, that irrespective of the actual plasmonic or dielectric cavity design, optimum cavity performance is limited to a finite spectral band. This actually emphasizes the importance of a spectrally narrow distribution of possible quantum emitter wavelengths as they necessarily must be matched to limited spectral bandwidths of cavities designed to modify the homogeneous emission linewidth for improved photon indistinguishability. We have added this information in the **revised manuscript on pages 12/13**.

More importantly though, the E₁₁*⁻ defects have a deeper optical trap depth than E₁₁* defects making them stable single photon emitters at room temperature. They also have a higher PLQY compared to E₁₁* defects in the same type of SWNTs (Berger *et al.*, *ACS Nano* **2019**, *13*, 9259-9269). A functionalization method that produces both types of defects or indeed mainly E₁₁* defects makes the reliable production of single photon emitters more challenging, simply because many nanotubes in a sample will not have the required properties. For fundamental studies this may not be a problem, however, it is one for practical applications.

3. The definition of the separate QYs for the E11*- and E11 is inappropriate. Excitons generated in the E11 state could have two routes to give out photons: either emit in the pristine segments or diffuse to the defects and emit at the E11*- energy. By the definition of QY, for certain absorbance, it is only meaningful to count the total emission from both the E11 and E11*- states. The authors should adjust the discussion of the E11 and E11*- peaks in Fig. 2d to PL intensity rather than separate QYs.

RESPONSE: We thank the Reviewer for pointing out this aspect and adjusted the discussion on **page 9 and labeling in Fig. 2d and Supplementary Fig. 15b** accordingly. We highlight the total PLQY and the spectral share (contribution) of the two emission features differently.

4. page 12: the signatures in the E₁₁ region has been solely assigned to localization of excitons. Previous studies have indicated that these are associated with certain sp³ configurations (e.g. ACS Nano 2017, 11, 10785). The authors should consider expanding their discussion on this.

RESPONSE: It is certainly true that some sp³-defect configurations are expected to show emission in the displayed E₁₁ region, as reported by He *et al.* (ACS Nano **2017**, *11*, 10785). Nevertheless, Kadria-Vili *et al.* (*J. Phys. Chem. C* **2016**, *120*, 23898–23904) showed that even untreated, “pristine” SWNTs can show emission features in this wavelength range due to unintentional defects introduced during the initial nanotube dispersion/sorting process. Thus, the structure of these defects and the step during which they were created remains elusive at this point. However, we agree with the reviewer that signatures in the E₁₁ region cannot be solely assigned to the localization of excitons and extended our discussion on this issue **on page 13 of the revised manuscript and caption of Supplementary Figure 12.**

REVIEWER COMMENTS

Reviewer #1 (Remarks to the Author):

I am satisfied with the provided corrections. Therefore, I would like to recommend this work for publication.

Kind regards,

Dawid Janas

Reviewer #2 (Remarks to the Author):

The authors mostly addressed my comments raised in the first review. Thus, after additional minor revision for the following comments, I recommend this paper to be accepted in this journal.

Regarding the response for comment #2 (reasons for the use of normalized PL spectra), I understand the three aspects including technical issues and effects of absorption spectral changes as problems to use the measured (non-normalized) PL spectra in figures. These are quite important to allow the readers to conduct reproducible experiments based on this technique. Therefore, the three aspects that the authors raised in their response should be described in the Experimental method section in the Supporting Information.

Regarding the response for comment #9 (characterization of attached functional groups), when the authors use the XPS datum as strong evidence for functionalization of the 2-fluoroaniline, more precise peak energy assignment of the F1s (such as C-F bond) is useful, which provides further structural information of the attached functional groups on the nanotubes by their functionalization technique.

Regarding the response for comment #12 (oxidation of the negatively charged SWNT intermediate), the authors raised moisture as an example of other effects, which quenches the base to reduce functionalization degrees. But, as another effect to reduce the functionalization degrees, reactions

of the generated carbanion with oxygen might be considered. Thus, this possibility should be also discussed.

Regarding the response for comment #14 (spectral intensity mismatch of E11*-), I understand that the authors need to use different SWNTs with different functionalization degrees for experiments in dispersion and film states. However, the authors' response is a little bit confusing because it is unclear whether the term of "Supplementary Figure 8a" in line 3 is for original or for revised manuscript. Considering from the following sentence of "the temperature-dependent change in defect emission is significantly smaller for E11*- defects compared E11* defects", it would be Supplementary Figure 9a in the revised manuscript. If so, the authors should have used highly functionalized SWNTs for the film experiments for Supplementary Figure 9a to ensure sufficient signal from E11*- defects. However, the revised caption in Supplementary Figure 11a in the revised manuscript says "Note that the same functionalized SWNTs were used to prepare the film in Fig. S9", which would be inconsistent with authors' explanation.

Moreover, the discussion why the intensity of E11*- defects decreased in the film state is missing in the manuscript.

In addition, Wang et al. reported that detrapping energies of the trapped excitons changed depending on the degree of functionalization (J. Phys. Chem. C 2016, 120, 11268–11276). Therefore, this point should be discussed because different functionalization degrees of the SWNTs were used for the experiments estimating detrapping energies.

Reviewer #3 (Remarks to the Author):

The authors have addressed the comments thoroughly.

Point-by-point Response

Manuscript # NCOMMS-20-46170A

Reviewer #1 (Remarks to the Author):

I am satisfied with the provided corrections. Therefore, I would like to recommend this work for publication.

RESPONSE: We thank the reviewer for their positive assessment.

Reviewer #2 (Remarks to the Author):

The authors mostly addressed my comments raised in the first review. Thus, after additional minor revision for the following comments, I recommend this paper to be accepted in this journal.

RESPONSE: We thank the reviewer and have addressed the remaining concerns as follows below.

Regarding the response for comment #2 (reasons for the use of normalized PL spectra), I understand the three aspects including technical issues and effects of absorption spectral changes as problems to use the measured (non-normalized) PL spectra in figures. These are quite important to allow the readers to conduct reproducible experiments based on this technique. Therefore, the three aspects that the authors raised in their response should be described in the Experimental method section in the Supporting Information.

RESPONSE: We agree with the reviewer and added this information to the Methods section in the revised Supporting Information (pages 3/4).

Regarding the response for comment #9 (characterization of attached functional groups), when the authors use the XPS datum as strong evidence for functionalization of the 2-fluoroaniline, more precise peak energy assignment of the F1s (such as C-F bond) is useful, which provides further structural information of the attached functional groups on the nanotubes by their functionalization technique.

RESPONSE: The observed binding energy of 689.5 eV for the F1s peak for 2-fluoroaniline-functionalized (6,5)-SWNTs is in good agreement with previously reported values for C-F bonds (see e.g. *J. Phys. Chem. C* **2013**, 117, 26166 and *Phys. Chem. Chem. Phys.* **2019**, 21, 10992) and thus further supports our proposed mechanism. We note, however, that precise structural assignments on the basis of absolute binding energies from XPS data are often complicated by charging effects that cannot be excluded for a thin film of semiconducting SWNTs with attached organic molecules even when deposited on a conductive substrate. We added this information and

references to the revised manuscript (page 16) and included further information about potential charging effects in the revised caption of Supplementary Figure 16.

Regarding the response for comment #12 (oxidation of the negatively charged SWNT intermediate), the authors raised moisture as an example of other effects, which quenches the base to reduce functionalization degrees. But, as another effect to reduce the functionalization degrees, reactions of the generated carbanion with oxygen might be considered. Thus, this possibility should be also discussed.

RESPONSE: We assume the reviewer refers to the reactive (e.g., aniline) carbanion intermediate because the possibility of oxidation of the SWNT carbanion has already been discussed in the manuscript.

We agree with the reviewer that oxidation of the generated reactive carbanions can represent a potential reason for the reduced degree of functionalization under atmospheric conditions. However, as discussed in the Supplementary Information we observed a strong enhancement of the degree of functionalization when DMSO was present as a co-solvent. Thus, we think that oxidation of the formed dimethyl anions to dimethylsulfone and methanesulfonic acid are more likely than oxidation of the reactive carbanion intermediate.

We have included both possibilities in our revised manuscript as in both cases a reduced degree of functionalization can be expected. Furthermore, we extended the discussion of this topic in Supplementary Information Figure 18.

Regarding the response for comment #14 (spectral intensity mismatch of E11*-), I understand that the authors need to use different SWNTs with different functionalization degrees for experiments in dispersion and film states. However, the authors' response is a little bit confusing because it is unclear whether the term of "Supplementary Figure 8a" in line 3 is for original or for revised manuscript. Considering from the following sentence of "the temperature-dependent change in defect emission is significantly smaller for E11*- defects compared E11* defects", it would be Supplementary Figure 9a in the revised manuscript. If so, the authors should have used highly functionalized SWNTs for the film experiments for Supplementary Figure 9a to ensure sufficient signal from E11*- defects. However, the revised caption in Supplementary Figure 11a in the revised manuscript says "Note that the same functionalized SWNTs were used to prepare the film in Fig. S9", which would be inconsistent with authors' explanation.

Moreover, the discussion why the intensity of E11*- defects decreased in the film state is missing in the manuscript.

RESPONSE: The reference "Supplementary Figure 8a" in line 3 of the original response was with regard to the original manuscript and Supplementary Information. However, the Figure numbers had changed between the original and the revised Supplementary Information. **To avoid confusion, we will only refer to the revised manuscript in this response.**

We used different samples for the measurements of emission in Supplementary Figure 8 and 10 and the same samples for measurements of Supplementary Figure 9 and 11. The revised captions in Supplementary Figure 11a are correct.

Regarding the use of low E_{11}^{*-} defect density in Supplementary Figs. 9 and 11: Certainly, the use of more highly functionalized SWNTs for these measurements would have the benefit of increased signal intensity of E_{11}^{*-} emission, however, we chose a sample with lower defect density to ensure that the defect sites with different optical trap depths were well separated from each other and defect-defect interactions were unlikely. In addition, choosing the same sample for the experiments presented in Supplementary Figs. 9 and 11 allowed us to directly compare measurements in thin film and dispersion.

The relative reduction of the E_{11}^{*-} signal compared to the E_{11}^* signal can be explained simply by their stronger power dependence compared to E_{11}^* defects (see Supplementary Figure 6). This effect is dramatically enhanced when performing PL measurements in thin films as the excitation density of the laser spot is much higher compared to measurements in dispersions. We added the explanation for the reduced relative E_{11}^{*-} intensity to the Supplementary Information (see section on ‘Temperature Dependence of Defect State Photoluminescence’, page 15).

In addition, Wang et al. reported that **detrapping energies of the trapped excitons changed depending on the degree of functionalization** (J. Phys. Chem. C 2016, 120, 11268–11276). Therefore, this point should be discussed because different functionalization degrees of the SWNTs were used for the experiments estimating detrapping energies.

RESPONSE: As correctly mentioned, Wang *et al.* reported an increase of detrapping energy at higher defect densities. Hence, we compared detrapping energies extracted from the data in Supplementary Figs. 10 and 11 for E_{11}^{*-} defects. Here we observe a similar trend, the extracted detrapping energies for E_{11}^{*-} defects increase slightly from 23 to 27 meV. We thank the reviewer for pointing out this aspect and we extended the discussion on this topic in the revised Supplementary Information on page 16.

Reviewer #3 (Remarks to the Author):

The authors have addressed the comments thoroughly.

RESPONSE: We thank the reviewer for their positive assessment.

REVIEWERS' COMMENTS

Reviewer #2 (Remarks to the Author):

The authors have addressed all comments and, therefore, I recommend to publish this paper in the current form.

Point-by-point Response
Manuscript # NCOMMS-20-46170B

Reviewer #2 (Remarks to the Author):

The authors have addressed all comments and, therefore, I recommend to publish this paper in the current form.

RESPONSE: We thank the reviewer for their positive assessment.